

# Functional renormalization group for fermions on a one dimensional lattice at arbitrary filling

**Lucas Désoppi[1,2*], Nicolas Dupuis[1] and Claude Bourbonnais[2]**

**1** Sorbonne Université, CNRS, Laboratoire de Physique Théorique de la Matière Condensée, LPTMC, F-75005 Paris, France
**2** Regroupement Québécois sur les Matériaux de Pointe and Institut Quantique, Département de physique, Université de Sherbrooke, Sherbrooke, Québec, Canada, J1K-2R1

* lucas.desoppi@usherbrooke.ca

## Abstract

A formalism based on the fermionic functional-renormalization-group approach to interacting electron models defined on a lattice is presented. One-loop flow equations for the coupling constants and susceptibilities in the particle-particle and particle-hole channels are derived in weak-coupling conditions. It is shown that lattice effects manifest themselves through the curvature of the spectrum and the dependence of the coupling constants on momenta. This method is then applied to the one-dimensional extended Hubbard model; we thoroughly discuss the evolution of the phase diagram, and in particular the fate of the bond-centered charge-density-wave phase, as the system is doped away from half-filling. Our findings are compared to the predictions of the field-theory continuum limit and available numerical results.

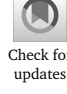

# 1   Introduction

The theory of interacting fermions in one spatial dimension gives the best understood examples of models whose asymptotic low-energy behavior distinctively deviates from that of a Fermi liquid, as commonly found in Fermi systems in higher dimension. Absence of quasi-particle excitations and power-law decay of correlation functions are governed by non-universal exponents characterized by very few hydrodynamic and interaction-dependent parameters which separate into spin and charge bosonic entities for spin-$\frac{1}{2}$ fermions [1–6]. Such distinctive features form the basis of the Luttinger liquid (LL) fixed-point phenomenology [7]. This is asymptotically accurate in the low-energy (continuum) limit, namely when the fermion spectrum can be considered as strictly linear around the Fermi points and when interactions projected on those points are considered as momentum independent. These are well known to be at the core of the field theory or continuum $g$-ology models of the interacting 1D Fermi gas. The fixed-point behavior of a linear LL proves to be generic for gapless branches of excitations of most models of interacting fermions in one dimension.

As one moves away in energy from the Fermi points the spectrum develops in practice some curvature. Deviations with respect to linearity alongside momentum dependence of interactions, although irrelevant in the renormalization group (RG) sense [7], were shown to modify the finite energy spectral properties predicted by the linear LL theory. Formulated in terms of an effective x-ray edge problem [8], the coupling of particles to a continuum of higher energy states is found to alter the power-law profiles of spectral lines near their absorption edges [9]. These non-linear LL effects could be rigorously checked in the case of integrable spinless-fermion models defined on a lattice [10–13].

Noticeable limitations of the linear $g$-ology mappings of non-integrable lattice models could also be found in the calculation of singular correlations that enter in the determination of their phase diagrams. This has been best exemplified in the case of the one-dimensional extended Hubbard model (EHM) for spin-$\frac{1}{2}$ fermions, which will serve here as the reference lattice model for the RG method developed in the present work.

At half-filling numerical calculations soon identified a shift of the continuous transition line connecting charge- and spin-density-wave states [14,15], a line that the continuum $g$-ology theory predicts to be gapless along the separatrix $U = 2V$, for the local ($U$) and nearest-neighbors ($V$) interaction parameters of the EHM. The origin of this alteration has resisted at least in weak coupling to all attempts of explanations formulated in the framework of the linear $g$-ology theory [16,17]. Using exact diagonalizations, Nakamura showed later on that the shift underlies the incursion of a distinct phase, known as a bond-centered charge-density-wave (BOW) phase. The BOW phase is entirely gapped in both spin and charge sectors and

extends across some finite region on both sides of the $U = 2V$ line of the phase diagram in weak coupling [18,19]. This was subsequently confirmed numerically both by quantum Monte Carlo [20, 21], and density-matrix RG methods [22–24].

On analytical grounds, Tsuchiizu and Furusaki showed from perturbation theory that by taking into account the momentum-dependent fermion-fermion scattering processes at high energy, that is, beyond the linear region, one can define, at some arbitrarily chosen lower energy, an effective weak-coupling linear $g$-ology model, but with modified and enlarged set of input parameters [25]. The modification is such that it allows the emergence of a BOW phase in the $U = 2V$ weak-coupling sector of the phase diagram [26,27]. Using a functional fermionic RG approach at the one-loop level, Tam *et al.* [28] pointed out that by integrating out numerically all the scattering processes for a discrete set of fermion momenta along the tight-binding spectrum in the Brillouin zone, the existence of a BOW phase can be found in the $U = 2V$ weak-coupling region of the EHM phase diagram at half-filling. Ménard *et al.* [29] thereafter formulated an RG transformation for half-filling tight-binding fermions in the Wilsonian scheme [30], in which irrelevant interaction terms can be classified from the momentum dependence of non-local scattering amplitudes away from the Fermi surface. Their impact on the low-energy RG flow has born out the presence of the BOW ordered phase where it is expected in the EHM phase diagram at weak coupling, alongside shifts of some other transitions lines where accidental symmetries are known to occur in the continuum $g$-ology limit.

These RG results focused on the EHM model at half-filling. It is the main motivation of the present work to extend away from half-filling the weak-coupling RG determination of quantum phases of 1D fermion lattice models in the presence of non local interaction. To achieve this program, we shall opt for the functional RG approach in the so-called one-particle-irreducible scheme [31, 32], which proves easier to implement analytically when dealing with the relatively unexplored situation of momentum dependent interactions and asymmetrically filled tight-binding spectrum. The one-loop RG flow equations for the momentum-dependent four-point vertices are expanded up to second order in the energy difference from the Fermi level for the asymmetric spectrum. The difference acts as the scaling variable which allows the power counting classification of marginal and irrelevant interaction terms, together with their interplay. The method developed below can in principle apply to any form of non-linear spectrum and momentum-dependent interactions in models with fermion density away from half-filling. From the calculations of the most singular susceptibilities the phase diagram of the EHM model can be mapped out. At half-filling the results confirm previous RG calculations for the existence of a gapped BOW phase near the $U = 2V$ line and bear out the shift of other transition lines between different ground states, in agreement with numerical results [19]. In both situations the role of the spectrum and irrelevant interactions terms in the qualitative change of initial conditions for an effective linear continuum theory in the low-energy limit can be established. The method is carried out away from half-filling and the region of dominant BOW gapped state is found to gradually shrink in size to ultimately be suppressed as a function of doping. The whole phase diagram then evolves towards an incommensurate situation but where noticeable modifications of the stability regions of quantum states, as predicted by the $g$-ology continuum model, are found. The integration of high-energy electronic states in the particle-hole-asymmetric non-linear part of the spectrum reveals the existence of screening effects coming from particle-particle pairing fluctuations which act at lower energy as an important factor in promoting singlet superconductivity or inversely either antiferromagnetism or triplet superconductivity against the charge-density-wave state.

The paper is organized as follows. In Sec. II the fRG method is introduced and the flow equations of couplings and various susceptibilities are derived at the one-loop level. In this framework known results of the EHM phase diagram in the limit of the continuum $g$-ology

model at and away from half-filling are recovered. In Sec. III, we broaden the formulation of fRG to include the tight-binding spectrum and the momentum-dependent interactions of the EHM, as actually defined on a lattice. The one-loop flow equations for marginal and up to second order for the set of irrelevant scattering amplitudes are derived. The phase diagrams at and away from half-filling are obtained and their comparison with the $g$-ology limit analyzed and critically discussed. We conclude this work in Sec. 4.

# 2 FRG for the extended Hubbard model at arbitrary filling

## 2.1 One-dimensional extended Hubbard model

The 1D extended Fermi-Hubbard model is defined by the Hamiltonian (in this paper, units are taken such that $k_B = \hbar = 1$ and the lattice constant $a = 1$)

$$\mathcal{H} = -t \sum_{i,\sigma} \left( c_{i,\sigma}^\dagger c_{i+1,\sigma} + \text{H.c.} \right) + U \sum_i n_{i,\uparrow} n_{i,\downarrow} + V \sum_i n_i n_{i+1}, \tag{1}$$

describing electrons moving on a lattice with a hopping amplitude $t > 0$ and experiencing on-site and nearest-neighbor interactions with strengths $U$ and $V$, respectively. In Eq. (1), $i$ denotes the site index, $\sigma = \uparrow, \downarrow$ is the spin index, $n_{i,\sigma} = c_{i,\sigma}^\dagger c_{i,\sigma}$ and $n_i = n_{i,\uparrow} + n_{i,\downarrow}$ is the number of electrons at site $i$.

The one-particle states have energies $\varepsilon(k) = -2t \cos(k)$ with wave vector $k$ of the tight-binding form, such that with respect to the Fermi level, these are comprised in the interval $-2t - \mu \leqslant \xi = \varepsilon - \mu \leqslant 2t - \mu$, where $\mu$ is the chemical potential. The tight-binding spectrum $\varepsilon(k)$ is shown in Fig. 1. The corresponding density of states is written as follows:

$$\mathcal{N}(\xi) = \frac{\Theta(2t - |\mu + \xi|)}{2\pi \sqrt{t^2 - (\xi + \mu)^2/4}}, \tag{2}$$

where $\Theta(x)$ is the Heavisde step function. It will indeed be useful to write the density of states for an arbitrary value of $\xi$, because in the RG flow, the momentum shell corresponding to the integration of the degrees of freedom will be taken at equal distance from the Fermi level for the empty and the occupied states (see Fig. 1). By definition, the Fermi level is related to the Fermi wave vector $k_F$, defined such that $\varepsilon(k_F) = \mu$. One can also define the Fermi velocity

$$v_F = \left. \frac{\partial \varepsilon}{\partial k} \right|_{k_F} = 2t \sin(k_F). \tag{3}$$

Let $n$ be the fermion filling number. Obviously, we have $0 \leqslant n \leqslant 2$. This number is directly given by an integration of the density of states up to the Fermi level:

$$n = 2 \int_{-\infty}^{0} d\xi \, \mathcal{N}(\xi) = 2 \int_{-k_F}^{+k_F} \frac{dk}{2\pi} = \frac{2k_F}{\pi}, \tag{4}$$

which leads to the simple relations:

$$k_F = \frac{\pi}{2} n, \qquad \mu = -2 \cos(\pi n/2), \tag{5}$$

where from now on $\mu$ is expressed in units of $t$.

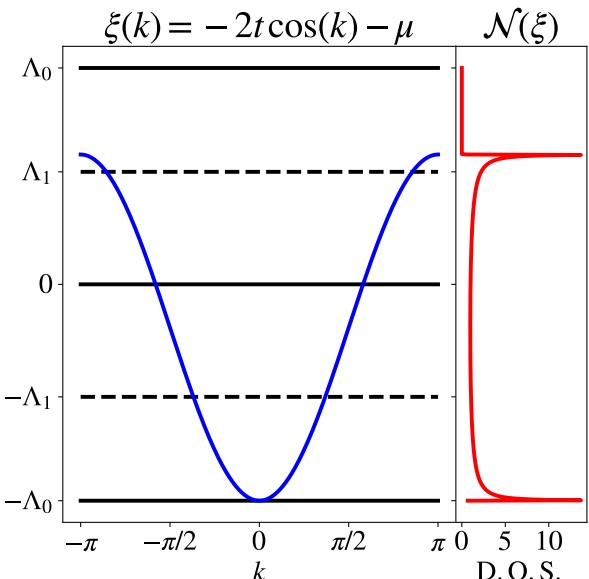

Figure 1: Tight-binding spectrum of the EHM model. Here $\Lambda_0$ is half of the initial bandwidth ($\Lambda_0 = 2t + |\mu|$) and $\Lambda_1$ is the energy cutoff at some intermediate step of the RG flow. On the right panel, $\mathcal{N}(\xi)$ the density of states as a function of energy showing the van-Hove singularities at the band edges.

In reciprocal space, the Hamiltonian of the EHM is written as (where $L$ denotes the number of lattice sites)

$$H = \sum_{k,\sigma} \left( \varepsilon(k) - U/2 \right) c^\dagger_{k,\sigma} c_{k,\sigma} + \frac{\pi v_F}{2L} \sum_{\{k,\sigma\}} g_{k_1,k_2,k'_1} c^\dagger_{k'_1,\sigma_1} c^\dagger_{k'_2,\sigma_2} c_{k_2,\sigma_2} c_{k_1,\sigma_1} \delta^{\mathrm{RL}}_{k_1+k_2-k'_1-k'_2}, \quad (6)$$

where $\delta^{\mathrm{RL}}$ denotes the momentum conservation condition on the lattice (RL stands for Reciprocal Lattice):

$$\delta^{\mathrm{RL}}_k = \sum_{n=-\infty}^{+\infty} \delta_{k,2\pi n}, \quad (7)$$

and the dimensionless coupling constants are given by:

$$g_{k_1,k_2,k'_1} = \frac{U}{\pi v_F} + \frac{2V}{\pi v_F} \cos(k_1 - k'_1). \quad (8)$$

## 2.2 One-loop flow equations

The EHM is studied with the functional RG. We first recast the partition function of the model into a field-theory setting at finite temperature $T = 1/\beta$, by means of a functional integral over a Grassmannian field $\varphi$:

$$\mathcal{Z} = \mathrm{Tr}\, e^{-\beta(\mathcal{H} - \mu \mathcal{N})} = \int \mathcal{D}[\varphi] e^{-\mathcal{S}[\varphi]}, \quad (9)$$

where the action $\mathcal{S}[\varphi]$ takes the form

$$\mathcal{S}[\varphi] = -\sum_{a',a,\sigma} \left[ G^0 \right]^{-1}_{a'a} \bar{\varphi}_{a',\sigma} \varphi_{a,\sigma} + \frac{T}{2L} \sum_{\{a',a,\sigma\}} V_{a'_1 a'_2 a_2 a_1} \bar{\varphi}_{a'_1,\sigma_1} \bar{\varphi}_{a'_2,\sigma_2} \varphi_{a_2,\sigma_2} \varphi_{a_1,\sigma_1}, \quad (10)$$

and the index $a \rightarrow (\omega_n, k)$ carries all the relevant information about momentum $k$ and fermionic Matsubara frequency $\omega_n = (2n+1)\pi T$.

The first term in the action is related to the free propagator $G^0$ which is diagonal in reciprocal space

$$\left[G^0\right]^{-1}_{a'a} = \left(\mathrm{i}\omega_n - \xi(k)\right)\delta_{a'a}. \tag{11}$$

The second term describes two-body interactions, and takes the following form:

$$V_{a'_1 a'_2 a_2 a_1} = \pi\nu_{\mathrm{F}} g_{k_1, k_2, k'_1} \delta^{\mathrm{RL}}_{k'_1 + k'_2 - k_2 - k_1} \delta_{\omega_{n'_1} + \omega_{n'_2} - \omega_{n_2} - \omega_{n_1}, 0}. \tag{12}$$

A quadratic term is added to the action,

$$\mathcal{S}[\varphi] \rightarrow \mathcal{S}[\varphi] + \sum_{a', a, \sigma} \bar{\varphi}_{a', \sigma} R_{\Lambda, a'a} \varphi_{a, \sigma}, \tag{13}$$

which regularizes the functional integral by suppressing the low-energy fluctuations. An anti-commuting source field $\eta, \bar{\eta}$ coupled to the fermion field is also included in the action which takes the form $\sum_{a, \sigma} \left(\bar{\eta}_{a, \sigma} \varphi_{a, \sigma} + \bar{\varphi}_{a, \sigma} \eta_{a, \sigma}\right)$. This gives the regularized generating functional of correlation functions $\mathcal{Z}_\Lambda[\eta]$. The regularized effective action $\Gamma_\Lambda[\phi]$ is then defined as the modified Legendre transform of the generating functional of connected correlation functions $\mathcal{W}_\Lambda[\eta] = \log \mathcal{Z}_\Lambda[\eta]$:

$$\Gamma_\Lambda[\phi] + \mathcal{W}_\Lambda[\eta] = \sum_{a, \sigma} \left(\bar{\eta}_{a, \sigma} \phi_{a, \sigma} + \bar{\phi}_{a, \sigma} \eta_{a, \sigma}\right) - \sum_{a', a, \sigma} \bar{\phi}_{a', \sigma} R_{\Lambda, a'a} \phi_{a, \sigma}, \tag{14}$$

where $\phi_{a, \sigma} = \langle \varphi_{a, \sigma} \rangle$ and $\bar{\phi}_{a, \sigma} = \langle \bar{\varphi}_{a, \sigma} \rangle$ with the expectation values computed in the presence of the source fields $\eta, \bar{\eta}$. The regularized effective action $\Gamma_\Lambda[\phi]$ satisfies the Wetterich equation [33–35]

$$\partial_\Lambda \Gamma_\Lambda[\phi] = \frac{1}{2} \mathrm{Tr} \left\{ \partial_\Lambda \boldsymbol{R}_\Lambda \left(\boldsymbol{\Gamma}^{(2)}_\Lambda[\phi] + \boldsymbol{R}_\Lambda\right)^{-1} \right\}, \tag{15}$$

where $\boldsymbol{\Gamma}^{(2)}_\Lambda[\phi]$ is the second functional derivative of the effective action with respect to the field. Additional source fields $J$ can be added to the effective action in order to generate flow equations for the response functions. The idea is then to decompose $\Gamma_\Lambda[\phi, J]$ as a sum of monomials $\Gamma^{[n,p]}_\Lambda[\phi, J] \sim \phi^n J^p$, and make identifications on both sides of the flow equation. We proceed at the one-loop level for which the 1-PI fRG hierarchy is truncated, this procedure leads to flow equations in weak coupling for the coupling constants $g$, three-leg vertices $Z$ and susceptibilities $\chi$. These equations have the familiar schematic form

$$\Lambda \partial_\Lambda g \sim \int \mathcal{L} g g, \quad \Lambda \partial_\Lambda Z \sim \int \mathcal{L} Z g, \quad \Lambda \partial_\Lambda \chi \sim \int \mathcal{L} Z Z, \tag{16}$$

where $\Lambda$ selects the degrees of freedom that are integrated at the step $\Lambda$. The corresponding one-loop diagrammatic contributions to the flow equations are shown in Figs. 2, 3 and 4, respectively, and where a simple line corresponds to the propagator $\boldsymbol{G}_\Lambda = \left(\boldsymbol{\Gamma}^{(2)}_\Lambda[\phi = 0] + \boldsymbol{R}_\Lambda\right)^{-1}$ and a slashed line to the single-scale propagator $\boldsymbol{S}_\Lambda = -\boldsymbol{G}_\Lambda \partial_\Lambda \boldsymbol{R}_\Lambda \boldsymbol{G}_\Lambda$.

## 2.3 Recovery of the $g$-ology continuum model

Before we take into account the non-linearity of the spectrum and the irrelevant coupling constants, it is useful for later comparisons to recover the well known $g$-ology electron gas model in the continuum limit, also known as the 1D electron gas model, for which lattice effects are mostly discarded. Thus, we linearize the tight-binding spectrum $\xi(k) = \varepsilon(k) - \mu$ in

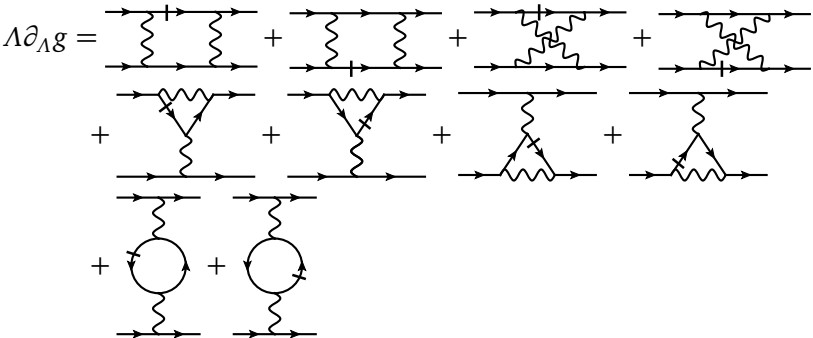

Figure 2: One-loop flow equations of the coupling constants in diagrammatic form. Here a slashed line refers to the single-scale propagator.

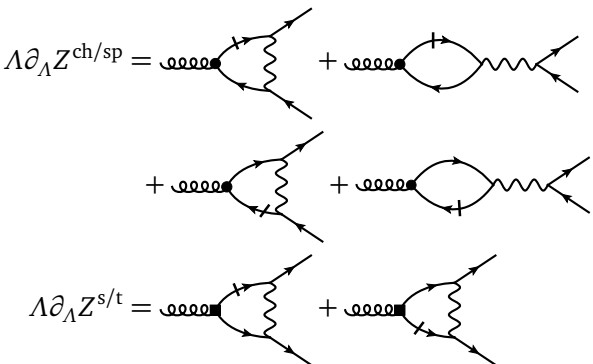

Figure 3: One-loop corrections to the flow equations of three-leg vertices for charge/spin-density-wave and singlet/triplet-pairing susceptibilities.

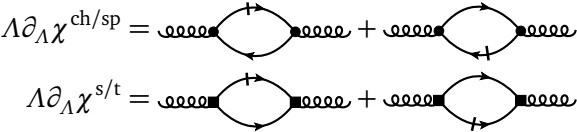

Figure 4: Flow equations for the charge/spin-density-wave and singlet/triplet-superconducting susceptibilities.

the vicinity of the two Fermi points $\pm k_\mathrm{F}$. We can write $k = \eta k_\mathrm{F} + (k - \eta k_\mathrm{F})$, where $\eta = \pm 1$ is the branch index, which gives

$$\xi(k) = \frac{\partial \varepsilon}{\partial k}\bigg|_{\eta k_\mathrm{F}} (k - \eta k_\mathrm{F}) + \ldots = v_\mathrm{F}(|k| - k_\mathrm{F}) + \ldots. \tag{17}$$

In the particular case of a half-filled band, $k_\mathrm{F} = \pi/2$. The reciprocal lattice wavevector $G = 2\pi$ being equal to $4k_\mathrm{F}$, we can identify $\pm 3k_\mathrm{F}$ with $\mp k_\mathrm{F}$. Away from half-filling, $G \neq 4k_\mathrm{F}$ and this identification does no longer hold. The $g$-ology model is obtained when the momenta

appearing in the coupling constants are evaluated on the Fermi points

$$
\begin{aligned}
g_1 &\equiv g_{+\eta k_{\mathrm{F}}, -\eta k_{\mathrm{F}}, -\eta k_{\mathrm{F}}} = \left(U - 2V(1 - \mu^2/2)\right)/\pi v_{\mathrm{F}}, \\
g_2 &\equiv g_{+\eta k_{\mathrm{F}}, -\eta k_{\mathrm{F}}, +\eta k_{\mathrm{F}}} = (U + 2V)/\pi v_{\mathrm{F}}, \\
g_3 &\equiv g_{+\eta k_{\mathrm{F}}, +\eta k_{\mathrm{F}}, -\eta k_{\mathrm{F}}} = \left(U - 2V(1 - \mu^2/2)\right)/\pi v_{\mathrm{F}}, \\
g_4 &\equiv g_{+\eta k_{\mathrm{F}}, +\eta k_{\mathrm{F}}, +\eta k_{\mathrm{F}}} = (U + 2V)/\pi v_{\mathrm{F}},
\end{aligned}
\tag{18}
$$

where the initialization condition (8) has been used. It appears that the constants $g_2$ and $g_4$ correspond to forward scattering, $g_1$ to backward scattering, while $g_3$ describes umklapp processes [36]. In order to find the expression of the two-particle vertex $\Gamma^{[4,0]}[\phi]$, we write its restriction close to the two Fermi points, which is indicated by the bracket $[\cdot]_{\mathrm{F}}$:

$$
\begin{aligned}
\left[\Gamma^{[4,0]}[\phi]\right]_{\mathrm{F}} = \frac{\pi v_{\mathrm{F}} T}{L} \sum_{\eta, \sigma_i} \sum_{\omega_{n_i}} \sum_{k_i \geqslant 0} \Bigg\{ & \frac{1}{2}(g_1 \delta_{\sigma_1, \sigma_3} \delta_{\sigma_2, \sigma_4} - g_2 \delta_{\sigma_1, \sigma_4} \delta_{\sigma_2, \sigma_3}) \\
& \times \bar{\phi}_{-\eta k_1', \sigma_3} \bar{\phi}_{+\eta k_2', \sigma_4} \phi_{-\eta k_2, \sigma_2} \phi_{+\eta k_1, \sigma_1} \\
& + \frac{g_3}{2} \bar{\phi}_{+\eta k_1', \sigma_1} \bar{\phi}_{+\eta k_2', \sigma_2} \phi_{-\eta k_2, \sigma_2} \phi_{-\eta k_1, \sigma_1} \\
& + \frac{g_4}{2} \bar{\phi}_{+\eta k_1', \sigma_1} \bar{\phi}_{+\eta k_2', \sigma_2} \phi_{+\eta k_2, \sigma_2} \phi_{+\eta k_1, \sigma_1} \Bigg\},
\end{aligned}
\tag{19}
$$

where momentum and Matsubara-frequency conservation is understood in the right-hand side.

When this simplified vertex is inserted in Eqs. (A.1) (see Appendix A.1), we end up with the well known $g$-ology flow equations

$$
\begin{aligned}
\Lambda \partial_\Lambda g_1 &= \mathcal{L}_{\mathrm{P}} g_1^2 - (\mathcal{L}_{\mathrm{C}} + \mathcal{L}_{\mathrm{P}}) g_1 g_2 - \mathcal{L}_{\mathrm{L}} g_1 g_4, \\
\Lambda \partial_\Lambda g_2 &= -\mathcal{L}_{\mathrm{C}} g_1^2/2 - (\mathcal{L}_{\mathrm{C}} + \mathcal{L}_{\mathrm{P}}) g_2^2/2 - \mathcal{L}_{\mathrm{P}'} g_3^2/2 - \mathcal{L}_{\mathrm{L}} g_4(g_1 - 2g_2), \\
\Lambda \partial_\Lambda g_3 &= (\mathcal{L}_{\mathrm{P}} + \mathcal{L}_{\mathrm{P}'}) g_3(g_1 - 2g_2)/2 - \mathcal{L}_{\mathrm{C}'} g_3(g_2 + g_4)/2, \\
\Lambda \partial_\Lambda g_4 &= -\mathcal{L}_{\mathrm{L}} (g_1^2 - 2g_2^2 + 2g_1 g_2 + g_4^2)/2 - \mathcal{L}_{\mathrm{C}'} (g_3^2 + g_4^2)/2.
\end{aligned}
\tag{20}
$$

Here the $\mathcal{L}_{\mathrm{X}}$'s are derivatives with respect to $\Lambda$ of the bubbles associated to particle-particle (p-p) and particle-hole (p-h) scattering channels in which C and C$'$ refer to inter- and intra-branch Cooper pairings, and P and L refer to Peierls and Landau channels. For this particular calculation a sharp cutoff is chosen (see appendix B concerning the regulator). This allows to compute the integrals in closed form and to recover the known results of the continuum limit which use a sharp cut-off procedure. The resulting bubbles can be classified into two logarithmically divergent bubbles of the p-p channel at zero momentum pair and the p-h one at momentum $2k_{\mathrm{F}}$, which leads to the most important contributions to the flow equations:

$$
\begin{aligned}
\mathcal{L}_{\mathrm{C}} &= \pi v_{\mathrm{F}} \Lambda \partial_\Lambda \int_{-\Lambda_0}^{\Lambda_0} \Theta(|\xi| - \Lambda) \, T \sum_{\omega_n} G^0(k_\xi, \omega_n) G^0(-k_\xi, -\omega_n) \, \mathrm{d}\xi, \\
\mathcal{L}_{\mathrm{P}} &= -\pi v_{\mathrm{F}} \Lambda \partial_\Lambda \int_{-\Lambda_0}^{\Lambda_0} \Theta(|\xi| - \Lambda) \, T \sum_{\omega_n} G^0(k_\xi - 2k_{\mathrm{F}}, \omega_n) G^0(k_\xi, \omega_n) \, \mathrm{d}\xi, \\
\mathcal{L}_{\mathrm{P}'} &= -\pi v_{\mathrm{F}} \Lambda \partial_\Lambda \int_{-\Lambda_0}^{\Lambda_0} \Theta(|\xi| - \Lambda) \, T \sum_{\omega_n} G^0(k_\xi + 2k_{\mathrm{F}}, \omega_n) G^0(k_\xi, \omega_n) \, \mathrm{d}\xi,
\end{aligned}
\tag{21}
$$

where $k_\xi = \arccos\left(-(\xi + \mu)/2\right)$. The last contribution $\mathcal{L}_{\mathrm{P}'}$ is affected by the fact that the nesting relation is not perfect away from half-filling. As a consequence, $\mathcal{L}_{\mathrm{P}}$ and $\mathcal{L}_{\mathrm{P}'}$ differ in

general, except at half-filling where $G = 4k_F$ [37]. The second category comes from non-divergent bubbles of p-p and p-h scattering channels (respectively noted $\mathcal{L}_{C'}$ and $\mathcal{L}_L$) when both particles belong to the same energy branch. These take the form

$$\mathcal{L}_{L,C'} = \mp \pi \nu_F \Lambda \partial_\Lambda \int_{-\Lambda_0}^{\Lambda_0} \Theta(|\xi| - \Lambda) \, T \sum_{\omega_n} G^0(k_\xi + 0^+, \omega_n) G^0(k_\xi, \omega_n) \, \mathrm{d}\xi \tag{22}$$

and only take finite values within the thermal shell $\Lambda \lesssim T$.

To determine the phase diagram, we have to derive further the flow equations for the three-leg vertices. This is done by adding to the effective action terms which couple the electronic field to the source field. One has to include terms associated to charge and spin density waves, centered on sites or on bonds, and singlet/triplet superconductivity:

$$\Gamma_Z[\phi, H, J] = \sum_{a,a',a''} \sum_{\sigma,\sigma'} \Big\{ Z^{\mathrm{ch}}_{aa',a''} \bar{\phi}_{a\sigma} H^0_{a''} \sigma^0_{\sigma\sigma'} \phi_{a'\sigma'} + Z^{\mathrm{sp}}_{aa',a''} \bar{\phi}_{a\sigma} \vec{H}_{a''} \cdot \vec{\sigma}_{\sigma\sigma'} \phi_{a'\sigma'}$$
$$+ Z^{\mathrm{s}}_{aa',a''} \bar{\phi}_{a\sigma} J^0_{a''} \pi^0_{\sigma\sigma'} \bar{\phi}_{a'\sigma'} + \mathrm{c.c.}$$
$$+ Z^{\mathrm{t}}_{aa',a''} \bar{\phi}_{a\sigma} \vec{J}_{a''} \cdot \vec{\pi}_{\sigma\sigma'} \bar{\phi}_{a'\sigma'} + \mathrm{c.c.} \Big\}, \tag{23}$$

where $\sigma^0$ is the $2 \times 2$ identity matrix, $\vec{\sigma} = (\sigma^1, \sigma^2, \sigma^3)$ is the vector containing the Pauli matrices, $\pi^0 = -i\sigma^2$ and $\vec{\pi} = -i\sigma^2\vec{\sigma}$. Furthermore, the $Z$ vertices have the following expressions:

$$Z^{\mathrm{ch-s}}_{a'a;a''} = Z^{\mathrm{ch-s}}_k(q) \delta^{\mathrm{RL}}_{k-k'+q} \delta_{\omega_n - \omega_{n'},0},$$
$$Z^{\mathrm{ch-b}}_{a'a;a''} = Z^{\mathrm{ch-b}}_k(q) \cos\big((k+q)/2\big) \delta^{\mathrm{RL}}_{k-k'+q} \delta_{\omega_n - \omega_{n'},0}, \tag{24}$$
$$Z^{\mathrm{s/t}}_{a'a;a''} = Z^{\mathrm{s/t}}_k(q) \delta^{\mathrm{RL}}_{k+k'+q} \delta_{\omega_n + \omega_{n'},0},$$

with the correspondences given for static source fields $H$ et $J$:

$$a \to (\omega_n, k), \quad a' \to (\omega_{n'}, k'), \quad a'' \to (0, q).$$

In the case of the $g$-ology model, we limit ourselves to the vertices $Z^{\mathrm{x}}_{\eta k_F}(q)$ evaluated at $q = \pm 2k_F$ for the density waves, and at $q = 0$ for the singlet/triplet superconductivity. There are four density waves at $2k_F$ which correspond to site-centered charge- and spin-density wave (CDW, SDW), and bond-centered charge- and spin-density wave (BOW, BSDW):

$$Z_{\mathrm{CDW}} = Z^{\mathrm{ch-s}}_{+k_F}(-2k_F) = Z^{\mathrm{ch-s}}_{-k_F}(+2k_F), \qquad Z'_{\mathrm{CDW}} = Z^{\mathrm{ch-s}}_{+k_F}(+2k_F) = Z^{\mathrm{ch-s}}_{-k_F}(-2k_F),$$
$$Z_{\mathrm{BOW}} = Z^{\mathrm{ch-b}}_{+k_F}(-2k_F) = Z^{\mathrm{ch-b}}_{-k_F}(+2k_F), \qquad Z'_{\mathrm{BOW}} = Z^{\mathrm{ch-b}}_{+k_F}(+2k_F) = Z^{\mathrm{ch-b}}_{-k_F}(-2k_F),$$
$$Z_{\mathrm{SDW}} = Z^{\mathrm{sp-s}}_{+k_F}(-2k_F) = Z^{\mathrm{sp-s}}_{-k_F}(+2k_F), \qquad Z'_{\mathrm{SDW}} = Z^{\mathrm{sp-s}}_{+k_F}(+2k_F) = Z^{\mathrm{sp-s}}_{-k_F}(-2k_F),$$
$$Z_{\mathrm{BSDW}} = Z^{\mathrm{sp-b}}_{+k_F}(-2k_F) = Z^{\mathrm{sp-b}}_{-k_F}(+2k_F), \qquad Z'_{\mathrm{BSDW}} = Z^{\mathrm{sp-b}}_{+k_F}(+2k_F) = Z^{\mathrm{sp-b}}_{-k_F}(-2k_F). \tag{25}$$

The vertices associated to singlet (SS) and triplet (TS) superconductivity are given by

$$Z_{\mathrm{SS}} = Z^{\mathrm{s}}_{+k_F}(0) + Z^{\mathrm{s}}_{-k_F}(0),$$
$$Z_{\mathrm{TS}} = Z^{\mathrm{t}}_{+k_F}(0) - Z^{\mathrm{t}}_{-k_F}(0). \tag{26}$$

The flow equations associated to the density-wave vertices are thus

$$\frac{\mathrm{d}Z_{\mathrm{x}}}{\mathrm{d}\ell} = \frac{1}{2} Z_{\mathrm{x}} \tilde{g}_{\mathrm{x}},$$
$$\tilde{g}_{\mathrm{CDW}} = (g_2 - 2g_1)\mathcal{L}_P - g_3\mathcal{L}_{P'}, \qquad \tilde{g}_{\mathrm{SDW}} = g_2\mathcal{L}_P + g_3\mathcal{L}_{P'}, \tag{27}$$
$$\tilde{g}_{\mathrm{BOW}} = (g_2 - 2g_1)\mathcal{L}_P + g_3\mathcal{L}_{P'}, \qquad \tilde{g}_{\mathrm{BSDW}} = g_2\mathcal{L}_P - g_3\mathcal{L}_{P'},$$

while those for singlet and triplet superconductivity are

$$\frac{\mathrm{d}Z_\mathrm{x}}{\mathrm{d}\ell} = \frac{1}{2}Z_\mathrm{x}\tilde{g}_\mathrm{x}\,,$$
$$\tilde{g}_\mathrm{SS} = (g_1 + g_2)\mathcal{L}_\mathrm{C}\,, \qquad \tilde{g}_\mathrm{TS} = (g_2 - g_1)\mathcal{L}_\mathrm{C}\,, \tag{28}$$

where $\ell$ is the so-called RG time, such that $\Lambda = \Lambda_0 \mathrm{e}^{-\ell}$. The initial conditions are $Z_\mathrm{x}(\ell = 0) = 1$ for all channels x.

The expression of the normalized susceptibility that stands for any of the above correlation channels is given by

$$\chi_\mathrm{x}(\ell) \quad = \int_0^\ell Z_\mathrm{x}^2(\ell')\left|\mathcal{L}_\mathrm{x}(\ell')\right|\mathrm{d}\ell'\,, \tag{29}$$

with $\chi_\mathrm{x}(\ell = 0) = 0$ as initial condition. The phase of the system is defined by the most singular susceptibility $\chi_\mathrm{x}$ and therefore the most singular $Z_\mathrm{x}$. We shall limit ourselves to the phases with the most important singularities. These correspond to $2k_\mathrm{F}$ density-wave and superconducting phases at zero pairing momentum, which are governed by Eqs. (27) and (28). The corresponding three-leg vertices can be expressed as $Z_\mathrm{x}(\ell) = \exp[\frac{1}{2}\gamma_\mathrm{x}(\ell)]$, with a scale-dependent exponent $\gamma_\mathrm{x}(\ell) = \int_0^\ell \tilde{g}_\mathrm{x}(\ell')\mathrm{d}\ell'$.

### 2.3.1 Half-filling

It is useful in what follows to recall the main features of the one-loop flow equations of the continuum theory both at and away from half-filling. We first consider the case at half-filling and in the zero-temperature limit where $\mu = 0$ ($n = 1$) and $\beta \to \infty$. This gives for the bubble intensities (21):

$$\mathcal{L}_\mathrm{P,P'} = -\mathcal{L}_\mathrm{C} = \tanh(\beta\Lambda/2) \to 1\,,$$
$$\mathcal{L}_\mathrm{L,C'} = \mp 2\Lambda\partial_\Lambda n_\mathrm{F}(\Lambda) \to 0\,, \tag{30}$$

where $n_\mathrm{F}$ is the Fermi distribution. From (20) one recovers the well known $g$-ology flow equations at half-filling [2, 36, 38]:

$$\frac{\mathrm{d}g_1}{\mathrm{d}\ell} = -g_1^2\,,$$
$$\frac{\mathrm{d}g_2}{\mathrm{d}\ell} = (g_3^2 - g_1^2)/2\,,$$
$$\frac{\mathrm{d}g_3}{\mathrm{d}\ell} = g_3(2g_2 - g_1)\,, \tag{31}$$
$$\frac{\mathrm{d}g_4}{\mathrm{d}\ell} = 0\,.$$

If the coupling constants remain weak for all values of $\ell$ then the electron system evolves towards a Tomanaga-Luttinger (TL) liquid with $g_2$ and $g_4$ couplings only and gapless excitations. On the other hand, if the flow of either $g_1$ or $(2g_2 - g_1, g_3)$ evolves towards a singularity at a critical $\ell_0$, the perturbative one-loop RG breaks down and we expect the formation of a gap $\Delta = \Lambda_0 \mathrm{e}^{-\ell_0}$ in the spin ($g_1 \to -\infty$) or charge ($2g_2 - g_1 \to +\infty, |g_3| \to +\infty$)[1] long-wavelength degrees of freedom.

The flow of $g_1(\ell)$ associated to the spin degrees of freedom is decoupled from those of $g_3(\ell)$ and $2g_2(\ell) - g_1(\ell)$ linked to the charge ones. These combine to give the scale invariant constant $C = g_3^2(\ell) - \left(2g_2(\ell) - g_1(\ell)\right)^2$ [36]. Thus for an initial attraction, $g_1 < 0$ ($U < 2V$),

---

[1]See also the footnote 2 below.

the flow of $g_1(\ell)$ scales to strong attractive coupling with a singularity that develops at a finite $\ell_\sigma$, indicative of a spin gap $\Delta_\sigma \sim \Lambda_0 e^{-\ell_\sigma}$; whereas for an initial repulsion $g_1 > 0$, $g_1(\ell)$ is marginally irrelevant and spin degrees of freedom remain gapless. For the charge part, when $g_1 - 2g_2 \geq |g_3|$, umklapp scattering becomes marginally irrelevant, $2g_2(\ell) - g_1(\ell)$ then scales to a non-universal value and the charge-density sector remains gapless. By contrast, when $g_1 - 2g_2 < |g_3|$, the umklapp term is marginally relevant and the flow leads to a singularity in both $g_3(\ell)$ and $2g_2(\ell) - g_1(\ell)$ at $\ell_\rho$ implying a Mott gap $\Delta_\rho \sim \Lambda_0 e^{-\ell_\rho} = \Lambda_0 e^{-1/\sqrt{|C|}}$ in the charge sector. Finally, at the one-loop level there are no logarithmic contributions to the flow of intra-branch forward scattering $g_4$, which remains scale invariant.

Regarding the phase diagram as a function of $U$ and $V$, when $U > |2V|$, so that $g_1 > 0$ and $g_1 - 2g_2 < |g_3|$, the strongest singularity appears for $\chi_{\text{SDW}}$, $\gamma_{\text{SDW}}$ being the largest exponent of (27), with a SDW state having gapless spin excitations and a Mott gap. For $V < -|U|/2$, so that $g_1 > 0$ and $g_1 - 2g_2 > |g_3|$, (27) yields $\gamma_{\text{TS}}$ as the largest exponent and a dominant susceptibility for TS with gapless excitations for both spin and charge. For $U/2 < V < 0$, which implies $g_1 < 0$ and $g_1 - 2g_2 > |g_3|$, it is in turn $\gamma_{\text{SS}}$ to be the largest exponent in (27) with a dominant singularity in the SS susceptibility with a spin gap. Finally when $U/2 < V$ and $V > 0$, we have $g_1 < 0$ and $g_1 - 2g_2 < |g_3|$ leading to a CDW phase, which is gapped for both spin and charge excitations. Along the separatrix $U = 2V$, $g_1 = g_3 = 0$, corresponding to gapless conditions of the TL model, $\gamma_{\text{SDW}} = \gamma_{\text{CDW}}$ and $\chi_{\text{CDW}}$ and $\chi_{\text{SDW}}$ are equally singular at $U > 0$, whereas at $U < 0$, $\gamma_{\text{SS}} = \gamma_{\text{TS}}$, and $\chi_{\text{TS}}$ and $\chi_{\text{SS}}$ are equal. Finally, the symmetry line at $U < 0$ and $V = 0$, with $g_1 < 0$ and $g_1 - 2g_2 > |g_3|$, leads to $\gamma_{\text{CDW}} = \gamma_{\text{SS}}$ and coexisting CDW and SS phases. The resulting well known phase diagram of the continuum theory is shown in Fig. 5 [38]. It is worth noting that in the $g$-ology model, $\chi_{\text{BOW}}$ never appears as the dominant susceptibility, but only as the subdominant one in the SDW phase [38].

### 2.3.2 Away from half-filling

We now turn to the main results for finite values of $\mu$. From (20), the one-loop flow equations at finite doping in the low-temperature limit can be put in the form

$$
\begin{aligned}
\frac{dg_1}{d\ell} &= -g_1^2 \,, \\
\frac{d}{d\ell}(2g_2 - g_1) &= g_3^2 \mathcal{L}_{P'} \,, \\
\frac{dg_3}{d\ell} &= (1 + \mathcal{L}_{P'})g_3(2g_2 - g_1)/2 \,, \\
\frac{dg_4}{d\ell} &= 0 \,.
\end{aligned}
\tag{32}
$$

These equations correspond to the former results of Seidel *et al.* [37, 39] and are consistent with those of the bosonization approach in the weak-coupling limit [6, 40, 41]. We illustrate this situation at a finite but small doping $\mu = 0.035$.

The flow of $g_1$, tied to the spin degrees of freedom, keeps the same form as before, namely $g_1(\ell) = g_1(1 + g_1\ell)^{-1}$, indicating a spin gap $\Delta_\sigma \sim \Lambda_0 e^{-1/|g_1|}$ when $g_1 < 0$ i.e. in the region above the separatrix $V = U/(2 - \mu^2)$ whose slope increases with $\mu$, as shown in Fig. 5-(b). Concerning the charge degrees of freedom, a finite $\mu$ affects the flows of $2g_2(\ell) - g_1(\ell)$ and $g_3(\ell)$ due to the suppression of the logarithmic singularity of the particle-hole loop $\mathcal{L}_{P'}$ when $\Lambda(\ell) < \mu$. Thus at sufficiently small couplings, the flow of $2g_2(\ell) - g_1(\ell)$ is no longer singular so that no charge gap is possible. This introduces gapless regions for the charge sector (Fig. 5-(b)) corresponding to either CDW or SDW phases. By cranking up $\mu$, the charge-gapped regions

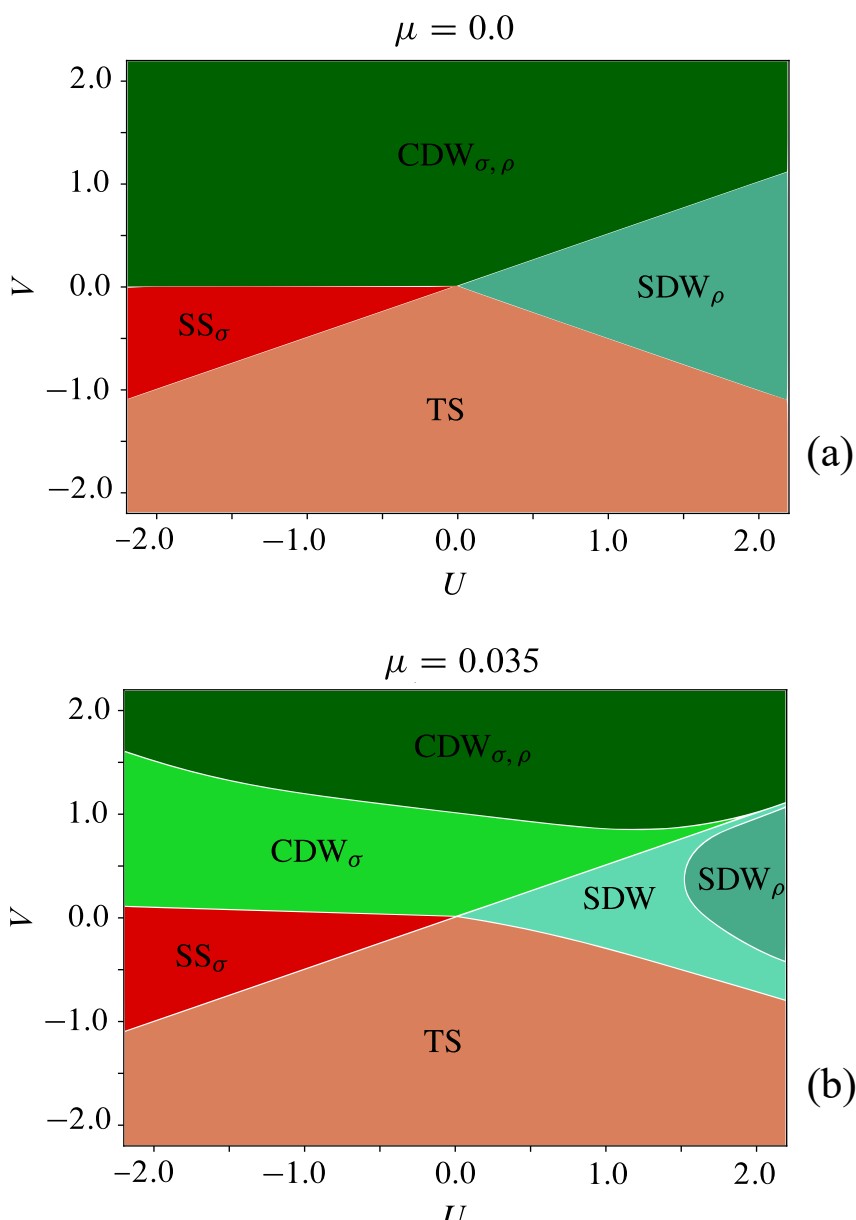

Figure 5: Weak-coupling phase diagram of the extended Hubbard model obtained from the $g$-ology model (continuum limit, linear spectrum and momentum-independent interactions) at half-filling $\mu = 0$ (a) and small doping $\mu = 0.035$ (b). $U$ and $V$ are expressed in units of bare hopping $t$. The subscripts $\sigma/\rho$ of a given phase indicate the presence of a gap in the spin/charge excitations (see also footnote 2).

persist[2] but shrink in size being pushed to higher couplings. In the gapless-charge domains, umklapp scattering reduces to a simple renormalization of the combination $2g_2 - g_1$, which becomes scale invariant in weak coupling [see Eq. (32)]. In the $g_1 > 0$ part of Fig. 5-(b), that

---

[2]The gap $\Delta_\rho$ that persists in the RG flow for some interval of $\mu \neq 0$ ($n \neq 1$) refers to the energy distance between the upper and lower Hubbard like sub-bands, which does not contract immediately to zero by doping away from half-filling. This finite excitation energy should not be confused with the Mott insulating gap which immediately closes at the metal-insulator transition when $n \to 1^{\pm}$. It is only at half-filling that this energy distance coincides with the insulating gap.

is for $V < U/(2 - \mu^2)$, the detrimental effect of doping on umklapp is also apparent for the gapless region where the most important power-law singularity in $\chi_{TS}$ gains in importance against SDW. A similar effect takes place at $g_1 < 0$ where the SS region, in which umklapp scattering is an irrelevant coupling, gains in importance against CDW when $\mu$ increases.

We shall examine next to what extent taking into account lattice effects of the EHM model can alter these results.

## 2.4 Lattice effects and low-energy limit

Lattice effects are twofold. First, they are present in the one-body term of the Hamiltonian through the inter-site hopping of electrons. This leads to the tight-binding spectrum of Fig. 1 showing the growth of its curvature as energy moves away from the Fermi level and becoming particle-hole asymmetric away from half-filling. Second, they appear in the coupling constants that are spatially non-local. This is the case of the nearest-neighbor interaction $V$ which introduces a dependence on wave vectors in momentum space.

Both effects are linked since the momentum dependence of interactions generates an additional curvature of the spectrum through one-particle self-energy corrections. At the one-loop level, these come from Hartree-Fock contributions to the flow. However, as shown in appendix A.3 those corrections are small in weak coupling and will therefore be ignored in the following.

We now turn to the effects of the lattice on coupling constants. When defined on the two Fermi points, like the $g$'s of the $g$-ology continuum theory considered above, they are known to be marginal. The momentum dependence of the coupling constants is irrelevant in the RG sense but can have both qualitative and quantitative effects on the phase diagram.

In order to classify the coupling constants, it is advantageous to consider the energy variables rather than the momenta [29, 30]. This is done using the dispersion $\xi(k)$ as measured with respect to the Fermi level. In fact, there is a one-to-one correspondence between momenta $k$ on the one hand, and $(\xi(k), \eta)$ on the other hand, where $\eta = \mathrm{sgn}\, k$ correspond to the branch index for positive $k$ ($\eta = +$) and negative $k$ ($\eta = -$). The idea is therefore to introduce a systematic expansion of the vertices and loops in power of the $\xi$ variables tied to the independent momenta $k$. For the couplings, one has

$$g_{k_1, k_2, k_1'} = g^{\vec{\eta}}(\vec{\xi}) = \sum_{n_i=0}^{\infty} \frac{\xi_1^{n_1} \xi_2^{n_2} \xi_{1'}^{n_1'}}{n_1! n_2! n_1'!} g_{\vec{n}}^{\vec{\eta}},$$

$$\vec{x} = (x_1, x_2, x_{1'}), \quad x = \xi, \eta, n.$$

(33)

where the coefficients $g_{\vec{n}}^{\vec{\eta}}$ now stand for the set of marginal and irrelevant interactions of the model.

We now derive the general form of the flow equations when the $\xi$ expansion is made explicit. One first makes the change of variables $k \rightarrow (\eta, \xi)$, and writes

$$\frac{\mathrm{d}g_{\vec{n}}^{\vec{\eta}}}{\mathrm{d}\ell} = -\Lambda \partial_\Lambda g_{\vec{n}}^{\vec{\eta}} = \sum_{\mathrm{x}} D_{\mathrm{x}}^{\vec{\eta}}(\vec{\xi}),$$

(34)

where the sum runs over all Feynman graphs of Fig. 2, that is to say $\mathrm{x} \in \{\mathrm{p, ph1, ph2, ph3}\}$. Furthermore, each diagram can be written in the form

$$D_{\mathrm{x}}^{\vec{\eta}}(\vec{\xi}) = \sum_p \mathcal{L}_{\mathrm{x}}^{\vec{\eta}}(p, \vec{\xi}) \gamma_{\mathrm{x}1}^{\vec{\eta}}(p, \vec{\xi}) \gamma_{\mathrm{x}2}^{\vec{\eta}}(p, \vec{\xi}),$$

(35)

where $\mathcal{L}_{\mathrm{x}}^{\vec{\eta}}(p, \vec{\xi})$ is a bubble of the scattering channel x, while the $\gamma_{\mathrm{x}i}^{\vec{\eta}}(p, \vec{\xi})$'s are combinations of the coupling constants. For example, in the case $\mathrm{x} = \mathrm{pp}$, one gets from Eqs. (A.3a) of

Appendix:

$$
\begin{aligned}
\mathcal{L}_{\mathrm{pp}}^{\vec{\eta}}(p,\vec{\xi}) &= \mathcal{L}_{p,-p+k_1+k_2}^{\mathrm{pp}}\,, \\
\gamma_{\mathrm{x}1}^{\vec{\eta}}(p,\vec{\xi}) &= g_{k_2,k_1,-p+k_1+k_2}\,, \\
\gamma_{\mathrm{x}2}^{\vec{\eta}}(p,\vec{\xi}) &= g_{p,-p+k_1+k_2,k_1'}\,.
\end{aligned}
$$

The corresponding expressions for the other channels x = ph1, ph2 and ph3 are given in Eqs. (A.3b), (A.3c) and (A.3d), respectively. It is then possible to make use of the expansion given in (33) for the couplings and a similar one for the bubbles. Once this is done, the flow equations are written as

$$
\sum_{n_i=0}^{\infty} \frac{\xi_1^{n_1}\xi_2^{n_2}\xi_{1'}^{n_1'}}{n_1!\,n_2!\,n_1'!}\frac{\mathrm{d}g_{\vec{n}}^{\vec{\eta}}}{\mathrm{d}\ell} = \sum_{n_i=0}^{\infty}\sum_{m_{1,i}=0}^{\infty}\sum_{m_{2,i}=0}^{\infty} \frac{\xi_1^{n_1}\xi_2^{n_2}\xi_{1'}^{n_1'}}{n_1!\,n_2!\,n_1'!}\mathcal{L}_{\vec{n},\vec{m}_1,\vec{m}_2}^{\vec{\eta},\vec{\eta}_1,\vec{\eta}_2}g_{\vec{m}_1}^{\vec{\eta}_1}g_{\vec{m}_2}^{\vec{\eta}_2}\,. \tag{36}
$$

Now it is useful to express the flow equations in a dimensionless form. Let us introduce the dimensionless quantities $\tilde{g}_\Lambda^{\vec{\eta}}(\tilde{\vec{\xi}})$, where $\tilde{\vec{\xi}} = \vec{\xi}/\Lambda$. The natural unit is the cut-off $\Lambda$,

$$
g_\Lambda^{\vec{\eta}}(\vec{\xi}) = \Lambda^{[g]}\tilde{g}_\Lambda^{\vec{\eta}}(\vec{\xi}/\Lambda) \iff \tilde{g}_\Lambda^{\vec{\eta}}(\tilde{\vec{\xi}}) = \Lambda^{-[g]}g_\Lambda^{\vec{\eta}}(\Lambda\tilde{\vec{\xi}})\,. \tag{37}
$$

In this expression, $[g]$ denotes the engineering dimension of the coupling constant $g$. For two-body interactions in one dimension, energy-independent coupling constants are dimensionless, $[g] = 0$. From the expansion (33), it is straightforward to determine the dimension of a generic coupling constant:

$$
g_{\vec{n}}^{\vec{\eta}} = \Lambda^{-|\vec{n}|}\tilde{g}_{\vec{n}}^{\vec{\eta}}\,, \tag{38}
$$

where the notation $|\vec{n}| = n_1 + n_2 + n_{1'}$ has been introduced. The dimensionless flow equations for the coupling constants are then obtained by a simple identification from (36):

$$
\frac{\mathrm{d}\tilde{g}_{\vec{n}}^{\vec{\eta}}}{\mathrm{d}\ell} = -\Lambda\partial_\Lambda\tilde{g}_{\vec{n}}^{\vec{\eta}} = -|\vec{n}|\tilde{g}_{\vec{n}}^{\vec{\eta}} - \sum_{n_{1,i}=0}^{\infty}\sum_{n_{2,i}=0}^{\infty}\tilde{\mathcal{L}}_{\vec{n},\vec{n}_1,\vec{n}_2}^{\vec{\eta},\vec{\eta}_1,\vec{\eta}_2}\tilde{g}_{\vec{n}_1}^{\vec{\eta}_1}\tilde{g}_{\vec{n}_2}^{\vec{\eta}_2}\,, \tag{39}
$$

with $\tilde{\mathcal{L}}_{\vec{n},\vec{n}_1,\vec{n}_2}^{\vec{\eta},\vec{\eta}_1,\vec{\eta}_2} = \Lambda^{|\vec{n}|-|\vec{n}_1|-|\vec{n}_2|}\mathcal{L}_{\vec{n},\vec{n}_1,\vec{n}_2}^{\vec{\eta},\vec{\eta}_1,\vec{\eta}_2}$. As a consequence, the expansion in $\xi$ classifies the coupling constants by order of irrelevance from the value of $|\vec{n}|$. In practice, we will restrict ourselves to quadratic order, i.e. $|\vec{n}| \le 2$. Let us also note that it would be possible to expand the vertices in power of the Matsubara frequencies [42,43]. It follows from dimensional analysis that the terms containing non-zero powers of the Matsubara frequencies are irrelevant. Since these terms are not present in the initial action (the interactions are not retarded), they can only be generated by the flow and are thus expected to remain negligible.

The different sets of interactions and their initial conditions can be expressed in terms of the coupling constants of the original EHM; see Eq. (8). The expansion of the cosine in terms of the variables $(\xi,\eta)$ gives, up to second order in $\xi$,

$$
\begin{aligned}
\cos(k_1-k_1') ={}& \eta_1\eta_{1'} + (1-\eta_1\eta_{1'})\frac{\mu^2}{4} + \frac{\mu}{4}(1-\eta_1\eta_{1'})(\xi_1+\xi_{1'}) \\
&- \eta_1\eta_{1'}\left(\frac{1}{8} - \frac{\mu^2}{32(1-\mu^2/4)}\right)(\xi_1^2 + \xi_{1'}^2) \\
&+ \left(\frac{1}{4} + \eta_1\eta_{1'}\frac{\mu^2}{16(1-\mu^2/4)}\right)\xi_1\xi_{1'} + \dots.
\end{aligned} \tag{40}
$$

Hence we obtain the following initialization conditions for the coupling constants introduced in (33). For marginal interactions $\mathcal{O}(\xi^0)$ ($\vec{n}=0$), one has

$$g_{0,0,0}^{+\eta,-\eta,-\eta} = g_1, \qquad g_{0,0,0}^{+\eta,-\eta,+\eta} = g_2,$$
$$g_{0,0,0}^{+\eta,+\eta,-\eta} = g_3, \qquad g_{0,0,0}^{+\eta,+\eta,+\eta} = g_4, \tag{41}$$

and the initial values coincide with those of the continuum theory in (18).

From (40) and at $\mathcal{O}(\xi)$ ($|\vec{n}|=1$), the set of irrelevant interactions labeled in terms of backward, forward and umklapp scattering amplitudes, together with their initial filling-dependent values, reads:

$$g_{1,0,0}^{+\eta,-\eta,-\eta} = \frac{V\mu}{\pi v_F}, \qquad g_{0,0,1}^{+\eta,-\eta,-\eta} = \frac{V\mu}{\pi v_F}, \qquad g_{1,0,0}^{+\eta,-\eta,+\eta} = 0, \qquad g_{0,0,1}^{+\eta,-\eta,+\eta} = 0,$$
$$g_{1,0,0}^{+\eta,+\eta,-\eta} = \frac{V\mu}{\pi v_F}, \qquad g_{0,0,1}^{+\eta,+\eta,-\eta} = \frac{V\mu}{\pi v_F}, \qquad g_{1,0,0}^{+\eta,+\eta,+\eta} = 0, \qquad g_{0,0,1}^{+\eta,+\eta,+\eta} = 0. \tag{42}$$

Likewise, the set of irrelevant couplings at $\mathcal{O}(\xi^2)$ ($|\vec{n}|=2$) and their initial values can be put in the form

$$g_{2,0,0}^{+\eta,-\eta,-\eta} = \frac{V}{\pi v_F}\left(\frac{1}{2} - \frac{\mu^2}{8(1-\mu^2/4)}\right), \qquad g_{0,0,2}^{+\eta,-\eta,-\eta} = \frac{V}{\pi v_F}\left(\frac{1}{2} - \frac{\mu^2}{8(1-\mu^2/4)}\right),$$
$$g_{1,0,1}^{+\eta,-\eta,-\eta} = \frac{V}{\pi v_F}\left(\frac{1}{2} - \frac{\mu^2}{8(1-\mu^2/4)}\right), \qquad g_{2,0,0}^{+\eta,-\eta,+\eta} = -\frac{V}{\pi v_F}\left(\frac{1}{2} - \frac{\mu^2}{8(1-\mu^2/4)}\right),$$
$$g_{0,0,2}^{+\eta,-\eta,+\eta} = -\frac{V}{\pi v_F}\left(\frac{1}{2} - \frac{\mu^2}{8(1-\mu^2/4)}\right), \qquad g_{1,0,1}^{+\eta,-\eta,+\eta} = \frac{V}{\pi v_F}\left(\frac{1}{2} + \frac{\mu^2}{8(1-\mu^2/4)}\right),$$
$$g_{2,0,0}^{+\eta,+\eta,-\eta} = \frac{V}{\pi v_F}\left(\frac{1}{2} - \frac{\mu^2}{8(1-\mu^2/4)}\right), \qquad g_{0,0,2}^{+\eta,+\eta,-\eta} = \frac{V}{\pi v_F}\left(\frac{1}{2} - \frac{\mu^2}{8(1-\mu^2/4)}\right),$$
$$g_{1,0,1}^{+\eta,+\eta,-\eta} = \frac{V}{\pi v_F}\left(\frac{1}{2} - \frac{\mu^2}{8(1-\mu^2/4)}\right), \qquad g_{2,0,0}^{+\eta,+\eta,+\eta} = -\frac{V}{\pi v_F}\left(\frac{1}{2} - \frac{\mu^2}{8(1-\mu^2/4)}\right),$$
$$g_{0,0,2}^{+\eta,+\eta,+\eta} = -\frac{V}{\pi v_F}\left(\frac{1}{2} - \frac{\mu^2}{8(1-\mu^2/4)}\right), \qquad g_{1,0,1}^{+\eta,+\eta,+\eta} = \frac{V}{\pi v_F}\left(\frac{1}{2} + \frac{\mu^2}{8(1-\mu^2/4)}\right). \tag{43}$$

The same expansion procedure can in principle be applied to the vertex parts of the response functions:

$$Z_k^X(q) = \sum_{n=0}^{\infty} \frac{\xi^n}{n!} Z_{\eta,n}^X(q). \tag{44}$$

However, at variance with the coupling constants, the irrelevant contributions to all $Z_X$ are zero at $\ell = 0$, so that their effect on the flow will be negligible. In the following, we shall therefore proceed to the evaluation of $Z_k^X(q)$ in the lowest or marginal order by retaining only $Z_{\eta,n=0}^X(q)$. Higher-order corrections are not expected to bring any qualitative modifications to the phase diagram. Thus for the site- and bond-density-wave channels at $q = \pm 2k_F$, the flow equations are respectively

$$\Lambda\partial_\Lambda Z_{CDW}' = \frac{1}{2}\mathcal{L}_{P'}(g_2 - 2g_1)Z_{CDW}' - \frac{1}{2}\mathcal{L}_P g_3 Z_{SDW},$$
$$\Lambda\partial_\Lambda Z_{CDW} = \frac{1}{2}\mathcal{L}_P(g_2 - 2g_1)Z_{CDW} - \frac{1}{2}\mathcal{L}_{P'}g_3 Z_{CDW}',$$
$$\Lambda\partial_\Lambda Z_{SDW}' = \frac{1}{2}\mathcal{L}_{P'}g_2 Z_{SDW}' + \frac{1}{2}\mathcal{L}_P g_3 Z_{SDW},$$
$$\Lambda\partial_\Lambda Z_{SDW} = \frac{1}{2}\mathcal{L}_P g_2 Z_{SDW} + \frac{1}{2}\mathcal{L}_{P'}g_3 Z_{SDW}', \tag{45}$$

and

$$\Lambda \partial_\Lambda Z'_{\text{BOW}} = \frac{1}{2} \mathcal{L}_{\text{P}'} (g_2 - 2g_1) Z'_{\text{BOW}} - \frac{1}{2} \frac{\mathcal{L}_{\text{P}}}{\cos(2k_{\text{F}})} g_3 Z_{\text{BOW}} ,$$

$$\Lambda \partial_\Lambda Z_{\text{BOW}} = \frac{1}{2} \mathcal{L}_{\text{P}} (g_2 - 2g_1) Z_{\text{BOW}} - \frac{1}{2} \mathcal{L}_{\text{P}'} \cos(2k_{\text{F}}) g_3 Z'_{\text{BOW}} ,$$

$$\Lambda \partial_\Lambda Z'_{\text{BSDW}} = \frac{1}{2} \mathcal{L}_{\text{P}'} g_2 Z'_{\text{BSDW}} + \frac{1}{2} \frac{\mathcal{L}_{\text{P}}}{\cos(2k_{\text{F}})} g_3 Z_{\text{BSDW}} ,$$

$$\Lambda \partial_\Lambda Z_{\text{BSDW}} = \frac{1}{2} \mathcal{L}_{\text{P}} g_2 Z_{\text{BSDW}} + \frac{1}{2} \mathcal{L}_{\text{P}'} \cos(2k_{\text{F}}) g_3 Z'_{\text{BSDW}} .$$

$$(46)$$

In the superconducting channel at zero pair momentum, one has

$$\Lambda \partial_\Lambda Z_{\text{SS}} = \frac{1}{2} \mathcal{L}_{\text{C}} (g_1 + g_2) Z_{\text{SS}} ,$$

$$\Lambda \partial_\Lambda Z_{\text{TS}} = -\frac{1}{2} \mathcal{L}_{\text{C}} (g_1 - g_2) Z_{\text{TS}} .$$

$$(47)$$

All the $Z_{\text{x}}$ equations are bound to the initial conditions $Z_{\text{x}}(\ell = 0) = 1$. From these the normalized susceptibilities $\chi_{\text{x}}$ in the channel x can be obtained from the definition (50) with initial condition $\chi_{\text{x}}(\ell = 0) = 0$. The main differences with respect to the $g$-ology model are in the bubbles and in the fact that the marginal coupling constants are influenced by the irrelevant ones.

We close this subsection by considering the uniform $q \to 0$ response for charge and spin densities corresponding to the uniform charge compressibility ($\chi_\rho$) and spin susceptibility ($\chi_\sigma$) whose divergences signal the occurrence of phase separation and ferromagnetism. In the framework of the fRG, these susceptibilities can be easily computed using the fact that the degrees of freedom contributing to the p-p and $2k_F$ p-h fluctuations come from non thermal energies $|\xi| \gtrsim T$. These are separated from those contributing to the $q \to 0$ response functions which rather correspond to the thermal width $|\xi| \lesssim T$. We then integrate first the flow equations (39) considering the Cooper and Peierls channels alone with $\Lambda$ running between $\Lambda_0$ and $T$. The renormalized marginal coupling constants thus obtained at the energy scale $T$ are then used to compute the uniform susceptibilities. Instead of integrating the flow with $\Lambda$ running between $T$ and 0, one can simply use an RPA, which is known to be exact for a linear spectrum (and equivalent to bosonization) once fluctuations due to back scattering and umklapp processes have been integrated out. For a sufficiently small temperature, the results effectively correspond to the $T = 0$ limit.

Explicitly one considers the uniform three-leg vertices which obey the flow equation

$$\frac{\text{d}Z_{\text{x}}}{\text{d}\ell} = \frac{1}{4} \mathcal{L}_{\text{L}} Z_{\text{x}} g_{\text{x}} ,$$

$$(48)$$

where

$$g_{\text{x}=\rho}(\ell) = g_1(\ell) - 2g_2(\ell) - g_4(\ell) ,$$

$$g_{\text{x}=\sigma} = g_1(\ell) + g_4(\ell) .$$

$$(49)$$

By inserting the one-loop RPA contributions to $g_{\text{x}}(\ell) = g_{\text{x}}^* / (1 - \frac{1}{2} g_{\text{x}}^* \chi_0(\ell))$ in the Landau scattering channel, one gets $Z_{\rho,\sigma}(\ell) = [1 - \frac{1}{2} g_{\rho,\sigma}^* \chi^0(\ell)]^{-1}$ where $\chi^0(\ell) = \frac{1}{2} \int_0^\ell \mathcal{L}_{\text{L}}(\ell') \text{d}\ell'$ which according to (30) gives a non-zero contribution when the integration $\ell'$ enters in the thermal energy interval ($\Lambda(\ell') \lesssim T$). Here $g_\rho^* = g_1^* - 2g_2^* - g_4$ and $g_\sigma^* = g_1^* + g_4$ and the starred $g_{1,2}^*$ couplings are the renormalized values obtained from (39) down to the edge of the thermal interval $\Lambda \sim T$. Following the definition of susceptibilities,

$$\chi_{\rho,\sigma}(\ell) \quad = \int_0^\ell Z_{\rho,\sigma}^2(\ell') \mathcal{L}_{\text{L}}(\ell') \text{d}\ell' ,$$

$$(50)$$

one gets the expression,

$$\chi_{\rho,\sigma}(T \to 0) = \frac{2}{1 - \frac{1}{2}g_{\rho,\sigma}^*}, \tag{51}$$

for the normalized uniform compressibility and spin susceptibility in the zero temperature limit. These coincides with the expressions derived by the functional integral method [44].

## 3 Lattice model: Results and discussion

In this section, we will discuss the consequences of lattice effects coming from the non-linearity of the spectrum and the momentum dependence of interactions in the determination of quantum phases of the EHM as a function of filling. All calculations are carried out at the arbitrary chosen temperature $T = 10^{-7}$ which regularizes the Fermi distribution functions while being consistent with the zero temperature limit. The calculations are also limited to the weak-coupling sector.

### 3.1 Half-filled case

Before considering non-zero values of the chemical potential, let us examine as a benchmark of our method the extensively studied half-filling case. The tight-binding spectrum at $\mu = 0$ shows a non-vanishing curvature as one moves away from the Fermi points $\pm k_F$. On the boundaries and at the center of the Brillouin zone, the spectrum displays a vanishing slope, which causes the appearance of a van Hove singularity. At half-filling the progressive integration of degrees of freedom is then symmetric with respect to occupied and empty states.

From the integration of Eqs. (39) and (45-47), and by using the intial conditions (18) and (42-43) for the couplings, one obtains the half-filling EHM phase diagram shown in Fig. 6. Among the most striking modifications with respect to the continuum $g$-ology phase diagram of Fig. 5-(a), we first note the phases located in the vicinity of the line $U = 2V$. Recall that in the $g$-ology framework, both $g_1$ and $g_3$ vanish along that line at half-filling, which leads to the conditions of the TL model; crossing the line then corresponds to a change of sign of the $g_1$ and $g_3$ coupling constants (see Eq. (18)).

Along the line $U = 2V > 0$ in the repulsive part of the diagram, the gapless regime of the TL model with equally singular SDW and CDW suceptibilities is made unstable by the presence of irrelevant couplings. Thus below but close to the line $U = 2V$, at the point C' in the phase diagram of Fig. 6, $g_3$ evolves to positive values and then becomes relevant together with the combination $2g_2 - g_1$. Both diverge at some critical $\ell_\rho$, indicative of a charge (Mott) gap. The fate of $g_1$ is of particular interest since though repulsive initially, it evolves toward negative values and its flow ultimately separates from those of $g_3$ and $2g_2 - g_1$ at sufficiently large $\ell$ where the influence of irrelevant terms in (39) at $|n| \neq 0$ becomes vanishinghly small and can be ignored above some arbitrary value $\ell^*$ or equivalently below an effective cutoff energy $\Lambda^* = \Lambda e^{-\ell^*}$. One finally recovers the flow of the continuum-limit theory [Eq. (31)], implying

$$g_1(\ell) = \frac{g_1(\ell^*)}{1 + g_1(\ell^*)(\ell - \ell^*)} \qquad (\ell \geq \ell^*), \tag{52}$$

where $g_1(\ell^*) < 0$. Typically, we have $|g_1(\ell^*)| \ll 1$, so that the singularity of (52) will invariably lead to a finite, though very small, gap $\Delta_\sigma \sim \Lambda^* e^{-1/|g_1(\ell^*)|}$ in the spin sector. Slightly above the $U = 2V$ line at the point C in the phase diagram, both $g_1$ and $g_3$ are initially attractive. While $g_1$ remains attractive and evolves to strong coupling with the formation of a spin gap $\Delta_\sigma$, which is much stronger in comparison to C', the coupling $g_3$, though initially attractive, changes sign and becomes repulsive at the beginning of the flow due to its coupling to irrelevant terms.

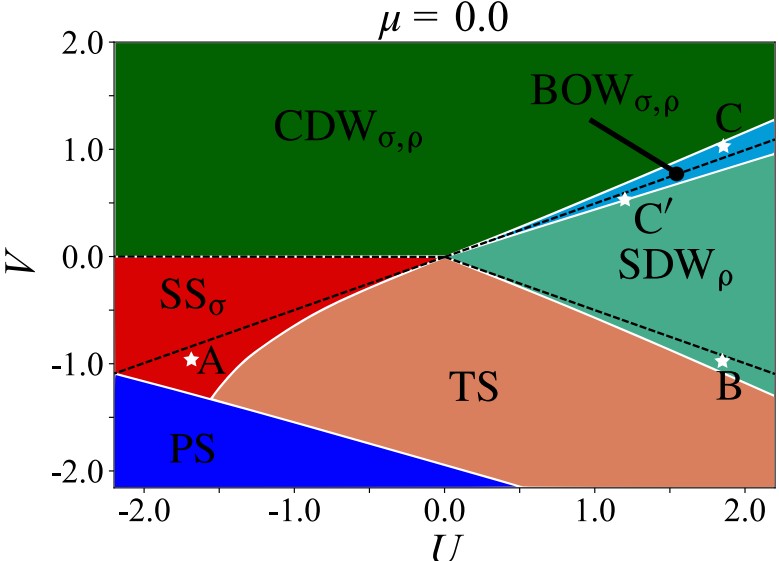

Figure 6: Phase diagram of the EHM at half filling. The points A, B, C and C' are discussed in the text. The dashed lines correspond to the phase boundaries of the continuum limit of the model shown in Fig. 5-(a).

According to Fig. 7-(b), the flows of $g_3$ and $2g_2 - g_1$ then evolve to strong coupling and lead to the formation of a charge gap $\Delta_\rho$.

The consequence of effective repulsive $g_3$ and attractive $g_1$ couplings on the nature of correlations is significant. On the $U = 2V$ line, instead of the coexistence of gapless CDW and SDW phases predicted by the TL model, a spin and charge gapped BOW phase emerges. According to Figs. 7-(a),(c), the gapped BOW state extends on either side of the line defining a fan shape region where it dominates over SDW and CDW phases. These findings confirm previous RG results [26, 28, 29, 45], and are consistent with those of numerical simulations in the weak-coupling region of the phase diagram [19, 20, 24].

We now turn to the attractive sector surrounding the $U = 2V$ line, namely the region $U < 0$ in the phase diagram of Fig. 6. In the $g$-ology formulation of the EHM, the TL conditions $g_1 = g_3 = 0$ at $U = 2V$ will be also unstable due to the presence of irrelevant terms that couple spin and charge degrees of freedom at the beginning of the flow. Thus in spite of $g_3$ remaining irrelevant, $g_1(\ell)$ becomes negative for $\ell \geq \ell^*$, as shown in Fig. 8-(b); $\ell^*$ being large, this leads to a small spin gap $\Delta_\sigma$ [Eq. (52)]. As displayed in Fig. 6, this tips the balance in favor of SS as the most stable phase, impinging on the region of TS stability found in the continuum $g$-ology theory (Fig. 5). The resulting growth of the SS region against the gapless TS one leads to a convex SS-TS boundary in the phase diagram that is consistent with previous weak-coupling RG calculations [29] and exact diagonalization results of Nakamura [19].

A related bending of phase boundary is also found for the $U = -2V$ line separating the Mott SDW and gapless TS phases in the TL model. The SDW state is then favored against TS at $U > 0$ and $V < 0$. This is illustrated in Fig. 8-(c),(d) for the point B of Fig. 6 where $g_2$ changes sign at the beginning of the flow, so that umklapp scattering, known to be irrelevant on the $U = -2V$ line in the continuum $g$-ology theory, becomes marginally relevant with a small but finite charge gap in the presence of irrelevant terms, which enlarges the stability region of the Mott SDW state.

Regarding the rest of the phase diagram of Fig. 6, only quantitative changes in the flow of coupling constants result from the presence of irrelevant terms due to lattice effects. These

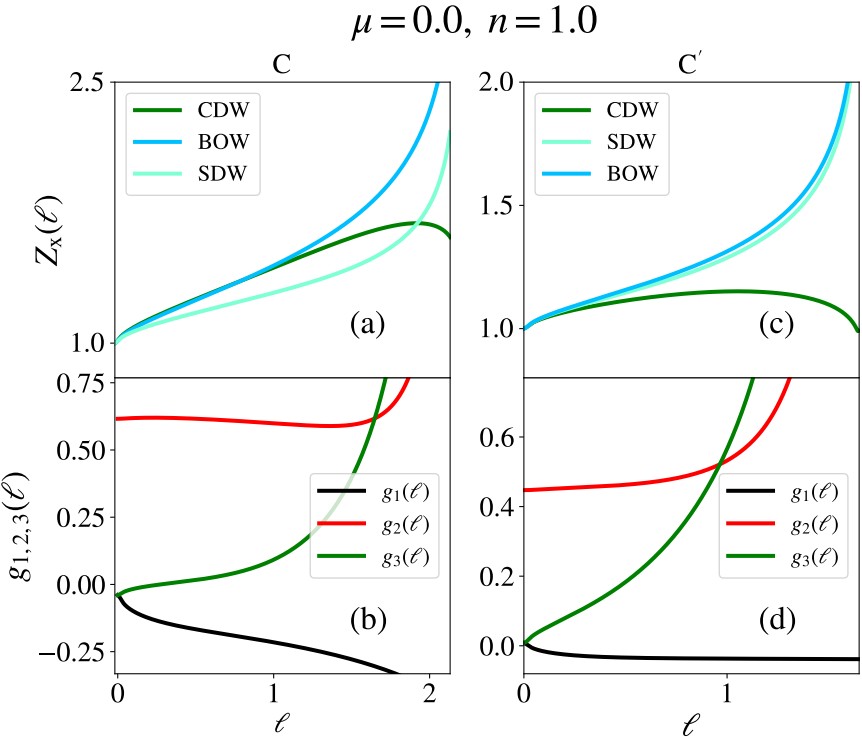

Figure 7: Flow of the three-leg vertices $Z_\text{x}$ of the susceptibilities [(a),(c)] and coupling constants [(b),(d)] for points C' and C of the phase diagram in Fig. 6, near the $U = 2V > 0$ line at half-filling. C:(1.81, 1.03), C':(1.43, 0.69).

results confirm those of Ref. [29] obtained by a different RG approach. We close the description of the phase diagram by pointing out the existence of a singularity in the uniform charge compressibility $\chi_\rho$. It signals an instability of the electron system against phase separation which makes an incursion in the zone of attractive $V$ in the phase diagram. This incursion is well established by numerical simulations [19, 46]. Note that for simplicity we didn't include in Fig. 6 and the following diagrams at different fillings the continuum prediction for phase separation.

We conclude that even if the lattice EHM model at half-filling is invariably described at sufficiently low energy by an effective continuum $g$-ology model, it is difficult to determine the initial conditions of this effective model without a careful analysis of the physics at high energy. Taking directly the continuum limit from the bare Hamiltonian may lead to wrong conclusions as to the nature of the ground state and in turn the structure of the phase diagram. These effects carry over away from half-filling for the EHM model, as we shall discuss next.

## 3.2 Away from half-filling

As far as the part played by the spectrum is concerned, we first note that away from half-filling, when $\mu \neq 0$, the integration of degrees of freedom is no longer symmetric with respect to the Fermi level, except in the low-energy domain where $\Lambda \ll \Lambda_0$ and the spectrum can be considered essentially linear, as generically depicted in Fig. 1.

As a consequence, the RG flow can be divided into three regimes. In the first regime, the asymmetry between electrons and holes plays an important role. Typically, for $\mu > 0$, we can have $\mathcal{N}(\xi > \Lambda) = 0$, that is, no fermion states are available, whereas $\mathcal{N}(\xi < -\Lambda) \neq 0$ (Fig. 1). The bubbles $\mathcal{L}^{\text{ph,pp}}$ will be affected accordingly. Thus, there will be no $2k_\text{F}$ particle-

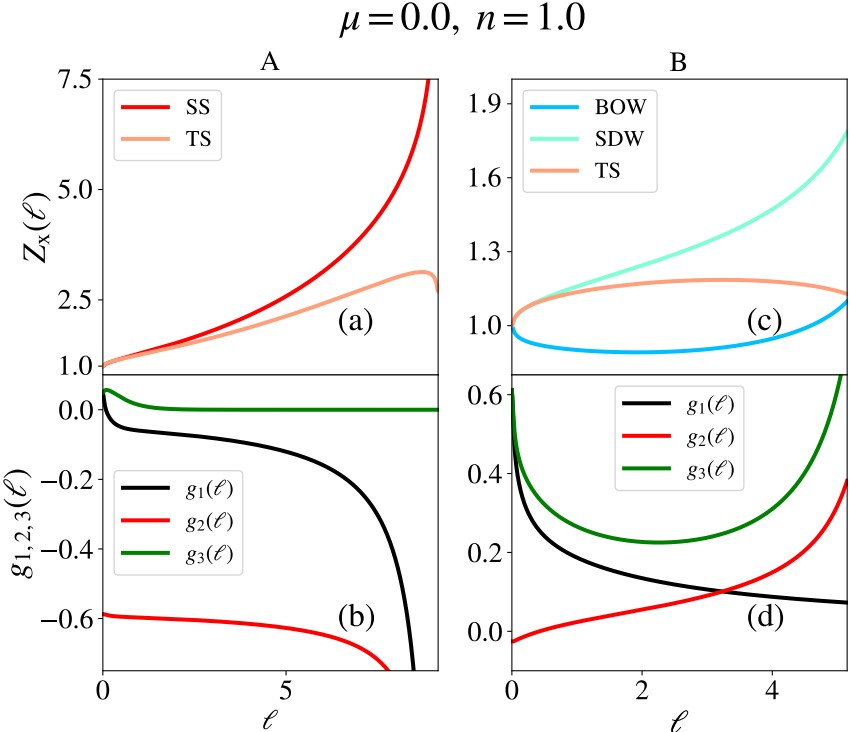

Figure 8: Flow of the three-leg vertices $Z_{\mathrm{x}}$ of the susceptibilities [(a),(c)] and coupling constants [(b),(d)] for points A and B of the phase diagram of Fig. 6 at half-filling. A:(-1.69, -1.0), B:(1,84, -1.0).

hole excitations and $\mathcal{L}^{\mathrm{ph}}$ will vanish in this regime (See Fig. 19). This contrasts with Cooper pair excitations, contributing to $\mathcal{L}^{\mathrm{pp}}$, which are present for $(-k, k)$ pairs of momentum where $\mathcal{N}(\xi_k) \neq 0$. It follows that $\mathcal{L}^{\mathrm{pp}}$ will be only halved in amplitude, the remaining part being still logarithmic. As we will see, this is responsible for a sizeable screening of interactions at the beginning of the flow, whose impact alters the structure of the phase diagram obtained in the continuum limit. This is reminiscent of the screening of Coulomb interactions by pairing fluctuations in the theory of conventional superconductivity [47]. The second regime corresponds to the $\Lambda$ range where we have $\mathcal{N}(\xi) \approx \mathcal{N}(-\xi)$, but where the spectrum is still poorly approximated by a linear function. In this regime, the logarithmic singularity of the p-h channel is only partly restored while the one in the p-p channel is complete (See Fig. 19); this imbalance between the two scattering channels favors the screening effects of the Coulomb term.

Finally, the last regime corresponds to the continuum limit at small $\Lambda$, for which we can write $\mathcal{N}(\xi) \approx \mathcal{N}(-\xi) \approx 1/\pi v_{\mathrm{F}}$. This corresponds to the density of states used in the $g$-ology model for each fermion branch and both spin orientations.

Besides these loop effects associated to the density of states, the chemical potential has also an impact on the Peierls loop $\mathcal{L}_{\mathrm{P}'}$ in which the reciprocal lattice vector is involved in momentum conservation making the nesting relation not perfect anymore [37,39]. As a consequence, $\mathcal{L}_{\mathrm{P}}$ survives but not $\mathcal{L}_{\mathrm{P}'}$, so that the equations for normal $g_1$ and $g_2$ processes become independent of umklapp processes at $\Lambda(\ell) < v_{\mathrm{F}}\mu$.

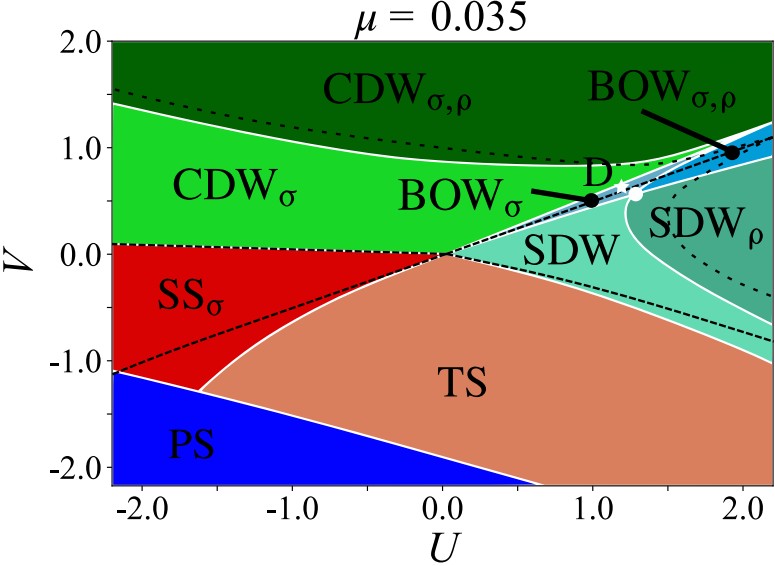

Figure 9: Same as Fig. 6 but away from half-filling: $\mu = 0.035$. The dashed lines refer to the phase boundaries of the continuum limit in Fig. 5-(b). The open circle corresponds to the threshold value $U_c(\mu)$ for the onset of a gapped BOW state as a function of $\mu$ (Fig. 10). The point D in the $BOW_\sigma$ charge-gapless region is discussed in the text and Fig. 11. The long-dashed lines indicate the boundary above which a charge gap is present in the continuum limit.

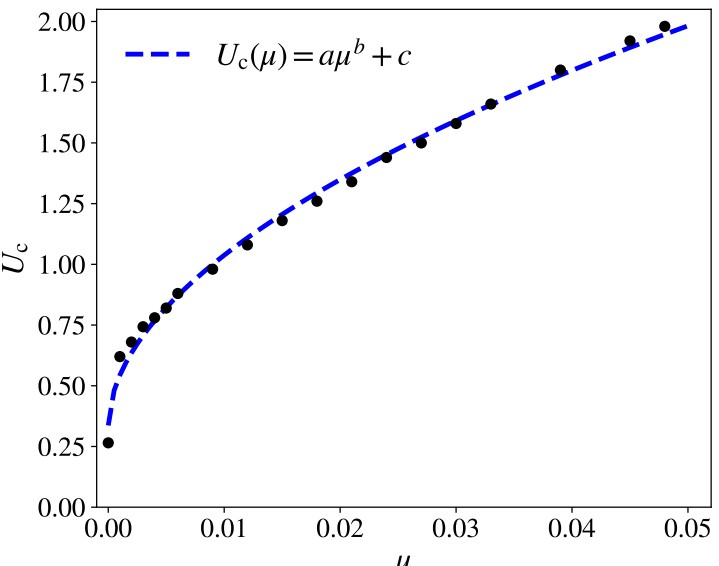

Figure 10: The critical coupling $U_c$ is plotted as a power law $U_c = a\mu^b + c$ of the chemical potential $\mu$. The gapped $BOW_{\sigma,\rho}$ phase exists for all $U \geqslant U_c$. Here $b = 0.53$, $a = 8.06$, and the constant $c = 0.34$ for a temperature of $10^{-7}$ used in the calculations.

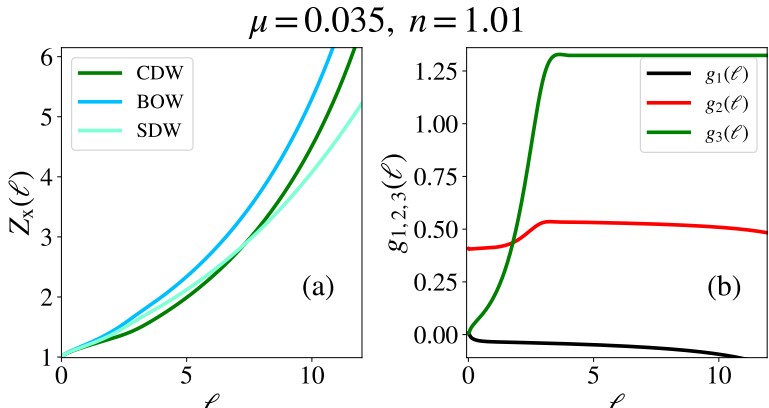

Figure 11: Flow of (a) the three-leg vertices $Z_X$ density-wave susceptibilities and (b) coupling constants at point D of the phase diagram of Fig. 9 ($\mu = 0.035$, $n = 1.01$). D:(1.30, 0.63).

### 3.2.1 Small doping

One can now consider the phase diagram for small departure from half-filling, namely at $\mu = 0.035$ (Fig. 9), integrating the flow equations (39), (45-46) and (47) with the initial conditions (18), (42) and (43).

In the repulsive sector near the $U = 2V$ line, we see that the regions with spin- and charge-gapped BOW and charge-gapped SDW phases shrink in size, and only exist above some threshold $U_c$ in the interactions. Thus a finite region unfolds at small coupling with CDW, BOW and SDW phases having no gap in the charge sector (see also footnote 2). The putative gap is indeed suppressed by the energy scale $v_F \mu$ that stops the flow of $2g_2 - g_1$ and $g_3$ towards strong coupling when $\Lambda(\ell) < v_F \mu$. The profile of the critical $U_c$ shown in Fig. 10 for the onset of the gapped BOW phase as a function of doping $\mu$, is well described by a power law $U_c(\mu) \simeq 8.03\mu^b + c$, where $b \simeq 0.53$. Here $c \to 0$ when the temperature goes to zero indicating that in the ground state, $U_c \to 0$ as $\mu \to 0$. At non-zero $\mu$, a finite region of dominant BOW state with only a spin gap and gapless charge excitations forms in the phase diagram. At point D in Fig. 9 for instance, the corresponding flow of the couplings displayed in Fig. 11 shows a growth followed by the leveling off of repulsive umklapp scattering. This is the signature that $g_3$ becomes irrelevant beyond some finite value of $\ell$. Nevertheless, this trajectory favors BOW correlations against CDW ones; it also initiates an incommensurate regime in which $2g_2(\ell) - g_1(\ell)$ evolves toward a constant. Regarding the attractive backscattering amplitude $g_1$, it will according to (52) invariably lead to a small spin gap at large $\ell$.

Dominant BOW correlations away from half-filling but at finite $U$ and $V$ near the line $U = 2V$ have been noticed numerically in quantum Monte Carlo simulations [20], in qualitative agreement with the present results. If one moves downward in the bottom right quadrant of the phase diagram of Fig. 9, we see that a finite $\mu$ suppresses the transition for the charge gap at the boundary between SDW and TS phases which is present at half-filling. The SDW phase then becomes entirely gapless near the boundary. As for the frontier between CDW and SS in top left quadrant of the phase diagram of Fig. 9, it has a bit moved upward which is consistent with the results of the continuum limit, as already shown in the lower panel of Fig. 5. However, as we will see next this boundary is noticeably affected at larger $\mu$.

One finally looks at the singularity line of the uniform charge compressibility $\chi_\rho$. In the attractive $V$ part of Fig. 9, the phase separation instability line calculated from (51) and the values of $g_\rho^*$ obtained at the end of the flow, undergoes only a small upward shift with respect to half-filled case.

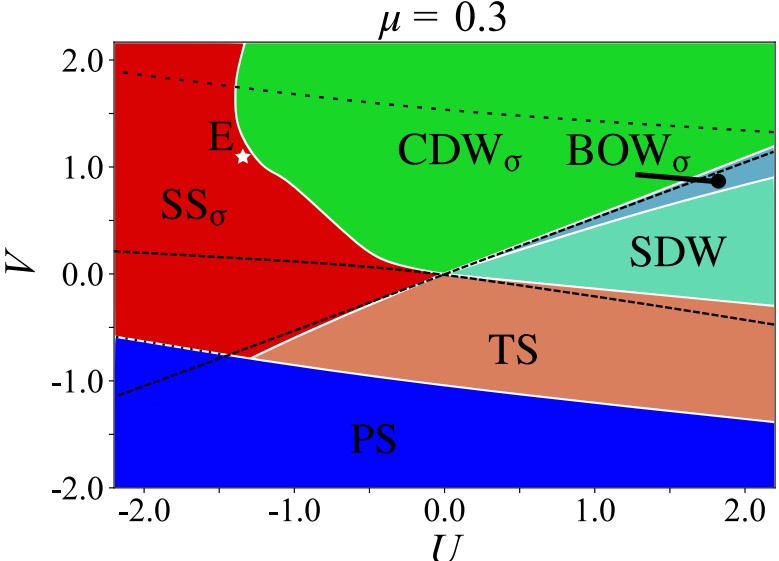

Figure 12: Same as Fig. 9 but for $\mu = 0.3$ ($n = 1.1$). The point E is discussed in detail in the text.

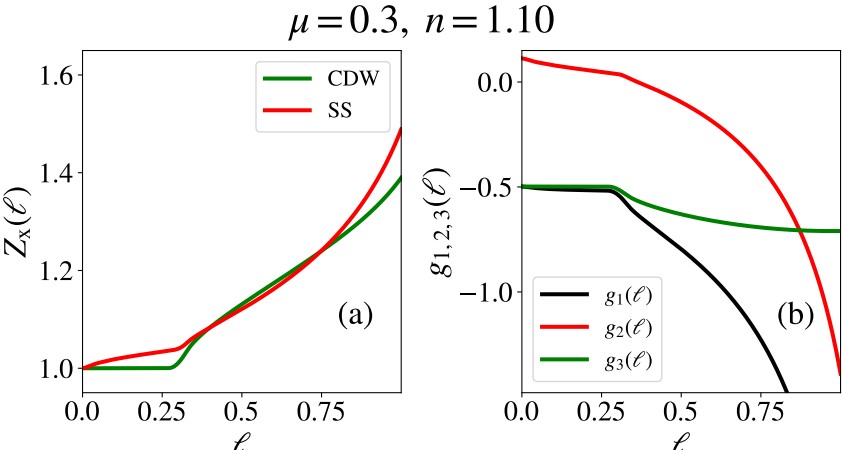

Figure 13: Flow of (a) the three-leg vertices $Z_x$ of CDW and SS susceptibilies and (b) the coupling constants at point E of the diagram in Fig. 12 ($\mu = 0.3$, $n = 1.10$). E:(-1.24, 0.97).

### 3.2.2 Intermediate doping

The phase diagram at intermediate doping $\mu = 0.3$ is displayed in Fig. 12. Due to the weak effect of umklapp processes at this filling, there is no region of the phase diagram characterized by a charge gap. However, the influence of $g_3$ at the beginning of the flow is still finite which, together with the change of $g_1$ to negative values due to irrelevant coupling terms, still defines near the $U = 2V$ line a region of dominant BOW phase at the incommensurate wave-vector $2k_F$. The characteristics of the flow of coupling constants in this BOW region, albeit much further reduced in their amplitudes, are similar to those shown in Fig. 11.

In the phase diagram of Fig. 12, the SDW-TS boundary turns out to be relatively close to the prediction of the model in the continuum limit. Here only the weak impact of umklapp and irrelevant couplings, which preserves the sign of $g_2$, restores the stability of TS compared

to the situation at very small $\mu$ (e.g., point B of Figs. 6 and 8-(c),(d)).

In the top left quadrant of Fig. 12 the deviations with respect to the prediction of the continuum model are particularly significative. One observes an expansion of the SS phase which goes well beyond its stability region found in the continuum limit; this occurs against CDW, which becomes secondary in importance. The origin of this expansion resides in the sizable asymmetry of the spectrum with respect to the Fermi level. At the beginning of the flow, that is at large $\Lambda$, all $2k_F$ particle-hole pair fluctuations coming from closed loops, vertex and ladder diagrams in Fig. 2 are strongly suppressed, a consequence of the lack of available density of states for either electrons or holes for this p-h pairing when asymmetry is pronounced, as illustrated in Fig. 1. This regime is followed by a second one at relatively large $\Lambda$ where these fluctuations are only partially restored. Thus there is a sizeable $\Lambda$ interval where p-p ladder diagrams for pairing fluctuations (first row of Fig. 2) dominate (see e.g. Fig. 19 at finite $\mu$), and govern the flow of $g_1$ and $g_2$. At point $E$ in Fig. 12 for instance, the coupling $g_2$, though initially repulsive, is screened by pairing fluctuations, to the point where it changes sign and becomes attractive. This is shown in Fig. 13-(b). As a result, the SS phase is favored against CDW (Fig. 13-(a)). This effect is reminiscent of the screening of the Coulomb interaction by pairing fluctuations which favors phonon-induced singlet superconductivity in isotropic metals [47]. The strong reduction of the $2k_F$ particle-hole pair contribution at the beginning of the flow is also responsible for making umklapp processes irrelevant in the whole CDW region of the upper half of the phase diagram. This is why no charge gap is found, in contrast to the continuum-limit prediction (region above the spaced dashed line in Fig. 12).

Finally we observe on Fig. 12 that the instability line for phase separation undergoes a sizable upward shift to weak coupling values. This is due to the smaller renormalization of $g_\rho^*$ coming from weaker umklapp scattering and, to a lesser extent, the decrease (increase) in Fermi velocity (density of states). Note that it has not been possible to extract with precision from the flow equations the $g_\rho^*$ value deep in the spin-gapped region of the lower left panel of the phase diagram (dashed white line of Fig. 12). In this region the energy $\Delta_\sigma \gg T$ at which the flow stops turns out to be far away from the thermal energy distance from the Fermi surface where $g_\rho^*$ is defined. In Fig. 12 and subsequent phase diagrams, the dashed line corresponds to an extrapolation of the line computed where the spin gap vanishes or is sufficiently close to the thermal scale.

### 3.2.3 Large doping

One now considers the phase diagram at the higher doping level, $\mu = 1.0$, shown in Fig 14. The whole diagram indicates that $g_3$ has virtually no effect in this range of doping reflecting an incommensurate situation for the electron system. This coupling can then be safely ignored in the analysis. Only a spin gap can occur. In the continuum model, we have seen that it is governed by the flow of $g_1(\ell)$ [Eq. (32)] and the initial condition $g_1 \simeq U - V < 0$ for attractive backward scattering, that is above the dashed line $U \simeq V$ in Fig. 14. According to the figure, the continuum-model result is however significantly altered by lattice effects and important deviations are present. A significant region develops with gapless spin excitations although $g_1$ is initially attractive. At point F for instance, Fig. 18 shows that $g_1$ indeed starts in the attractive domain but rapidly evolves towards repulsive sector to become a marginally irrelevant variable at sufficiently large $\ell$.

This remarkable effect has its origin in the pronounced asymmetry of the spectrum which, as we have seen, suppresses most, if not all, contributions coming from $2k_F$ particle-hole loops at large $\Lambda$; small momentum pairing fluctuations coming from contributions of ladder Cooper diagrams to $g_1$ in Fig. 2 largely dominate. Since for these terms the product $g_1 g_2$ in lowest order is initially negative, this makes these diagrams globally positive and pushes the flow of $g_1$ towards positive values. This can be seen as the counterpart effect of the screening discussed

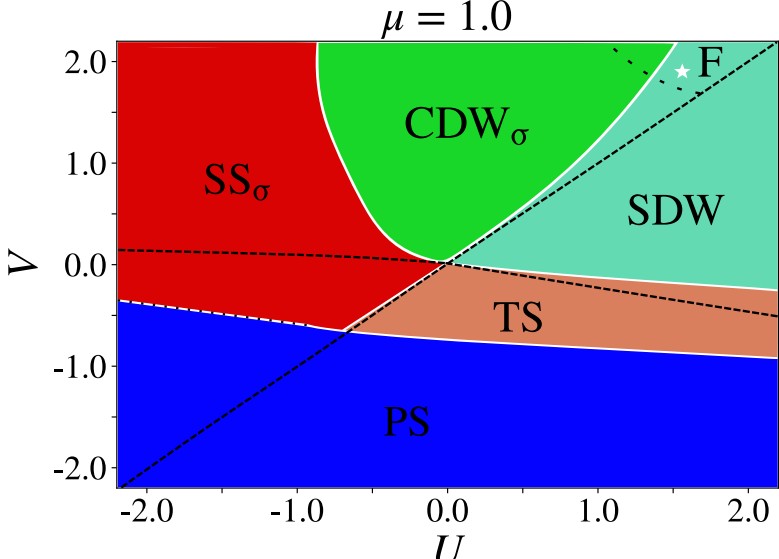

Figure 14: Same as Fig. 12 but for $\mu = 1.03$ ($n = 1.1$). The point F is discussed in the text and Fig. 18.

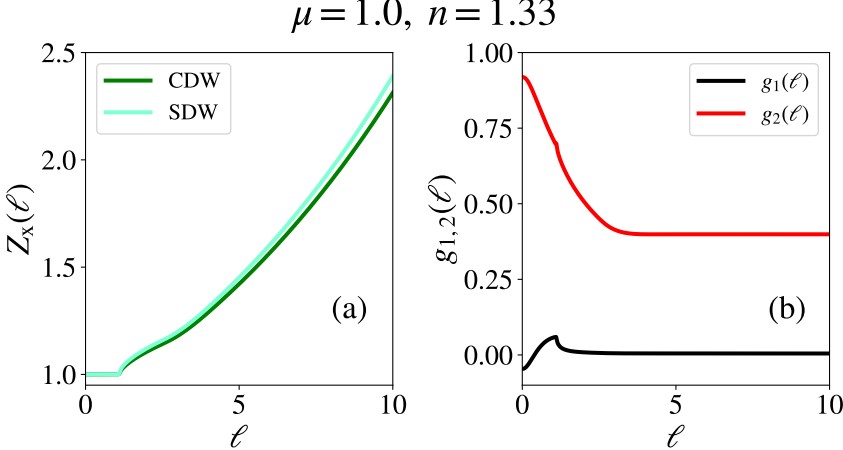

Figure 15: Flow of the three-leg vertices $Z_x$ of the susceptibilities (a) and couplings (b) at point F of Fig. 14 at $\mu = 1.0$ ($n = 1.33$). F:(1.50, 1.75).

above for the enhancement of singlet superconductivity by pairing fluctuations. This counter-screening of $g_1$ enlarges the region of gapless spin degrees of freedom in comparison with the continuum $g$-ology prediction. This in turn expands the SDW phase at the expense of the spin-gapped CDW phase whose correlations, though still singular, become secondary in importance, as shown in Fig. 18-(a). Exact diagonalization studies of the EHM carried out at $n = 2/3$ ($\mu = -1.0$) on the electron-doped side and which corresponds to $n = 4/3$ ($\mu = 1.0$) in the hole-doped case of Figs. (14-18), have clearly identified such corrections to the spin gap line of the continuum $g$-ology approach [48]. However, in this enlarged region with no spin gap, the superconducting TS and SS susceptibilities are not enhanced with respect to the free electron limit.

In the attractive $V$ region the phase separation line continues to be slightly shifted upward to weaker coupling due to more favorable initial values $g_\rho$ and lower $v_F$. As in the previous case with $\mu = 0.3$, the values of $g_\rho^*$ cannot be accurately extracted from the flow in the spin

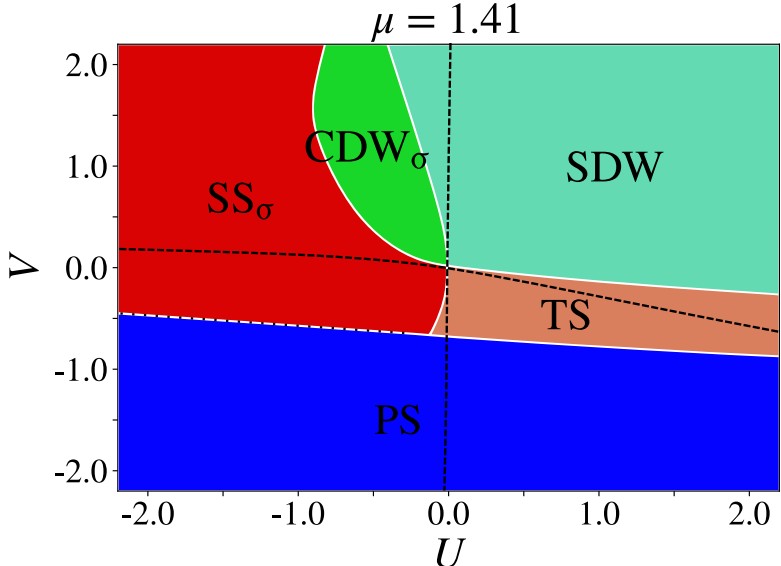

Figure 16: Phase diagram of the EHM at 3/4-filling $\mu = \sqrt{2}$ ($n = 1.5$).

gapped region (dashed white line of Fig. 16), where the flow stops at an energy scale far from $T$ and thus away from the conditions of the continuum limit.

If we turn our attention to the top left quadrant of the phase diagram, we see that compared to the results shown in Fig. 12 the stability region of SS phase is further broadened against the CDW one of smaller amplitude. The SS region reaches about twice the area predicted by the continuum model. The screening of $g_2$ by pairing fluctuations from positive to negative values results from the non-linear spectrum in the first two regimes of the flow. This follows the pattern already displayed in Fig. 13, which is here magnified due to the more pronounced asymmetry of the spectrum. This trend is confirmed when $\mu$ is further increased.

This is illustrated by the calculations performed at the higher commensurate doping $\mu = \sqrt{2}$ (3/4-filling, $n = 1.5$). These yield the phase diagram shown in Fig. 16 which is roughly similar to Fig. 14, except for the boundaries delimiting the CDW phase. By the same mechanism of screening the CDW region monotonously shrinks in size showing sign of closing at stronger coupling, this to the benefit of the SS or SDW phase. On the SDW side, this is concomitant with the expansion of gapless region for spin degrees of freedom due to counter screening at the beginning of the flow. Results of exact diagonalisations at quarter-filling are congruent with these corrections [48, 49].[3] Regarding the instability line of phase separation at attractive $V$, only a small upward shift in its position is found with respect with the previous $\mu = 1.0$ or $n = 1.3$ situation owing to the slight increase in the initial value of $g_\rho$ and of the density of states.

When the doping is further increased beyond the 3/4-filling, qualitative changes in the phase diagram become manifest. As shown in Fig. 17 for $\mu = 1.5$ ($n = 1.54$), the region of gapless spin degrees of freedom continues to be enlarged with respect to the one of the continuum $g$-ology results, but the most striking result resides in the closing of the CDW zone at $V > 0$ which gives way to the emergence of a TS phase with gapless excitations in the spin

---

[3]The present fRG calculations do not take into account the influence of the $8k_F$ umklapp scattering which involves the transfer of four particles from one Fermi point to the other and that is present at 3/4 (or 1/4) filling [6]. This higher-order umklapp scattering is $\mathcal{O}(\xi^2)$ in power counting and is thus irrelevant. It is known to only affect qualitatively the phase diagram beyond some critical $V > 0$ value where a $4k_F$ charge ordered state is found [50,51]. This regime is well outside the weak coupling sector considered in this work. However, one cannot exclude that it may affect the flow of marginal couplings.

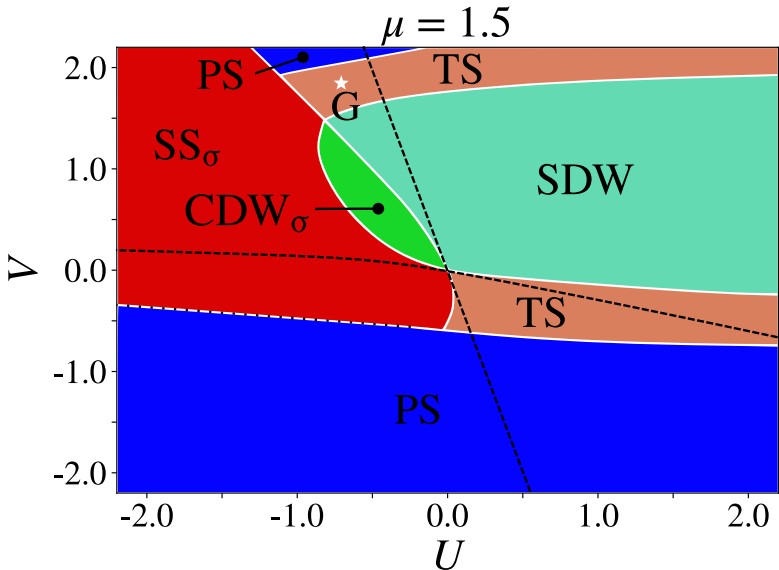

Figure 17: Phase diagram of the EHM at large $\mu = 1.5$ ($n = 1.54$).

sector. This result contrasts with what is found in the $g$-ology framework where the TS phase is confined to the lower right part of the phase diagram. As shown in Fig. 18-(b), the counter screening of $g_1$ ($g_2$) to positive (lower) values at the beginning of the flow is responsible for the occurrence of gapless superconductivity in this part of the phase diagram. As displayed in Fig. 18-(a), only TS correlations are singularly enhanced in this region while those of the SS type are reduced compared to the free electron limit.

To our knowledge no numerical simulations have been carried out at this doping which would allow a precise comparison. However, results of exact diagonalizations at quarter filling ($\mu = -1.41$) have revealed the existence of a peculiar and unexpected superconducting phase in the gapless sector for $V \gtrsim 4$ and $U$ nearly centered around zero [48, 49]. This would be located above the weak-coupling CDW region of Fig. 16. The present results strongly suggest that superconductivity found by exact diagonalizations at 1/4-filling corresponds to the TS phase that emerges in Fig. 17 at a smaller $V > 0$. Finally, the phase separation line in the attractive $V$ domain continues its slow upward shift, another instability line of this type begins to appear, but this time in the TS region described above at repulsive $V$. It is worth mentioning that at larger positive $V$ numerical simulations achieved at 1/4-and 2/3-fillings also find such a phase within the previously described TS region [48, 49].

From the above results one can conclude that the asymmetry between occupied and unoccupied electron states in an incommensurately filled spectrum can introduce pairing fluctuations which act as an efficient mechanism to modify the repulsive part of long-range Coulomb interactions at low energy and thus promote superconductivity of different nature.

## 4  Conclusions and perspectives

In this work we have developed a weak-coupling functional RG approach to 1D lattice models of interacting fermions in one dimension. In the framework of the EHM, we have shown how lattice effects modify in a systematic way the initial conditions defining the effective continuum field theory which invariably emerges at sufficiently low energy. For repulsive couplings at half-filling, for instance, the impact of irrelevant interactions on marginal couplings, which

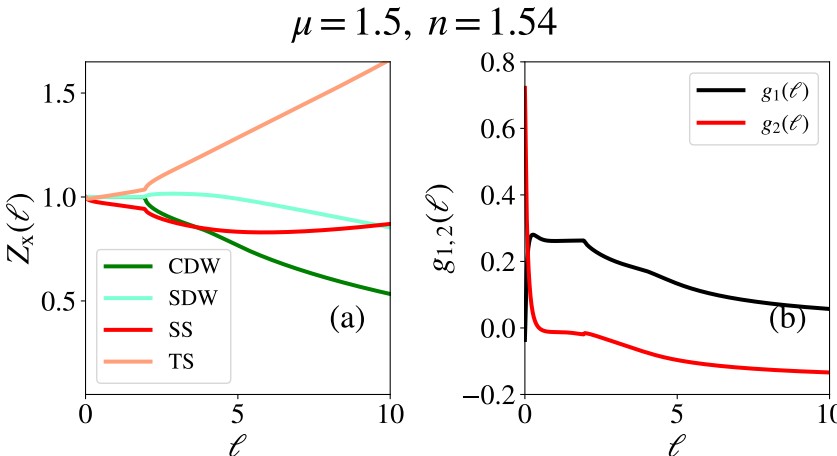

Figure 18: Flow of the three-leg vertices $Z_x$ of the susceptibilities (a) and couplings (b) at point G of Fig. 17 at $\mu = 1.5$ ($n = 1.54$). G:(-0.60, 1.80).

couple spin and charge degrees of freedom, turn out to be a key factor in the emergence of the gapped BOW state that overlaps the $U = 2V > 0$ gapless TL line of the continuum theory. We have also checked that qualitative changes in the nature of ground states are also manifest in the attractive sector of the EHM phase diagram at half-filling. These changes are due to irrelevant terms affecting the flow of marginal couplings at high energy and introducing noticeable shifts in the transition lines of the continuum theory, altering the stability region of the gapless TS state in favour of SS or SDW gapped states. These alterations of the continuum EHM phase diagram at weak coupling are consistent with previous numerical studies [18,19]; they also confirm the results obtained from numerical fRG in the repulsive coupling sector [28], and more generally from a Wilsonian RG approach to the non linearity of the spectrum and momentum-dependent interaction of the EHM [29].

We have also carried out our fRG procedure away from half-filling. In this case, the particle-hole symmetry in the tight-binding spectrum is lost and the integration of degrees of freedom becomes asymmetric with respect to the Fermi level. This notably affects the influence of high-energy fermion states on the flow of scattering amplitudes and susceptibilities. An imbalance between the logarithmic screening of the p-p and $2k_F$ p-h scattering channels is introduced which couples charge and spin degrees of freedom. In a finite energy interval at the beginning of the flow, the $2k_F$ density-wave part and concomitantly the magnification of umklapp commensurability, are strongly reduced. This contrasts with the p-p scattering channel which is weakly affected and sees its logarithmic singularity maintained. As the integration of degrees of freedom approaches the Fermi level, the imbalance together with irrelevant interactions scale down to zero and the flow progressively evolves toward the one of an effective continuum theory. However, the input parameters that govern the low-energy flow are not those of the naive continuum limit and alter sizable parts of the EHM phase diagram compared to the continuum $g$-ology predictions away from half-filling. This is particularly manifest for the CDW state whose extent in the phase diagram as the dominant phase at negative $U$ and repulsive $V$, for instance, is steadily reduced as a function of doping to the benefit of singlet superconductivity which gains in importance. This feature is not without bearing comparison with the screening of Coulomb interactions by high-energy pairing fluctuations in ordinary metals, which is known to promote the existence of superconductivity from retarded attractive coupling induced by electron-phonon interactions [47]. At large doping and repulsive $V$, pairing fluctuations are found to promote repulsive back scattering interactions by expanding the region of gapless spin excitations. This occurs with the emergence of a TS phase compat-

ible with the one found by exact diagonalization studies carried out at stronger coupling far from half-filling [48, 49].

The approach developed in this paper can be easily transposed to other non-integrable models of interacting electrons defined on a lattice. This is the case of models with generalized non-local interactions [52–55], for which numerical calculations are available at half-filling and known to deviate from the predictions of the g-ology approach in the field-theory continuum limit [19]. Another natural extension of the present work concerns the EHM in the quasi-one-dimensional case, where a weak but finite interchain hopping is taken into account. This may serve as a weak-coupling quasi-1D EHM to study the sequence of ground states that can unfold in strongly anisotropic correlated systems as a function of doping. Some of these applications are currently under investigation.

C. B and L. D thank the National Science and Engineering Research Council of Canada (NSERC), the Regroupement Québécois des Matériaux de Pointe (RQMP) and the Institut Quantique of Université de Sherbrooke for financial support. The authors thank E. Larouche and M. Haguier for their support on various numerical aspects of this work.

# A Flows of coupling constants

## A.1 Finite-temperature, one-dimensional, single-band systems

In this first part of the Appendix, we detail the derivation of the flow equations for the scattering amplitudes at the one-loop level for both marginal and irrelevant couplings. To do so we first make the correspondence $k \to (\eta, \xi)$ between the momentum and the energy $\xi$ and its branch $\eta$, so that

$$g_{k_1,k_2,k_1'} = g^{\vec{\eta}}(\vec{\xi}),$$

where $\vec{x} = (x_1, x_2, x_{1'})$ for $x = \xi, \eta$. From the diagrams of Fig. 2, the flow equations of the coupling constants at the one-loop level comprise a sum of contributions coming from p-p and p-h scattering channels, which can be put in the form:

$$\Lambda \partial_\Lambda g^{\vec{\eta}}(\vec{\xi}) = \sum_{\text{x}} D_{\text{x}}^{\vec{\eta}}(\vec{\xi})$$
$$= D_{\text{pp}}^{\vec{\eta}}(\vec{\xi}) + D_{\text{ph1}}^{\vec{\eta}}(\vec{\xi}) + D_{\text{ph2}}^{\vec{\eta}}(\vec{\xi}) + D_{\text{ph3}}^{\vec{\eta}}(\vec{\xi}), \tag{A.1}$$

where the diagrams

$$D_{\text{x}}^{\vec{\eta}}(\vec{\xi}) = \sum_p \mathcal{L}_{\text{x}}^{\vec{\eta}}(\vec{\xi}) \gamma_{\text{x1}}^{\vec{\eta}}(\vec{\xi}) \gamma_{\text{x2}}^{\vec{\eta}}(\vec{\xi}) \tag{A.2}$$

are expressed in terms of loops $\mathcal{L}_{\text{x}}^{\vec{\eta}}(\vec{\xi})$ and combinations of coupling constants $\gamma_{\text{x1}}^{\vec{\eta}}(\vec{\xi})$ and $\gamma_{\text{x2}}^{\vec{\eta}}(\vec{\xi})$ for each scattering channel x. They are respectively given by

$$D_{\text{pp}}^{\vec{\eta}}(\vec{\xi}) = \sum_p \mathcal{L}_{p,-p+k_1+k_2}^{\text{pp}} g_{k_2,k_1,-p+k_1+k_2} g_{p,-p+k_1+k_2,k_1'}, \tag{A.3a}$$

$$D_{\text{ph1}}^{\vec{\eta}}(\vec{\xi}) = \sum_p \mathcal{L}_{p,p-k_1'+k_2}^{\text{ph}} g_{k_1,p-k_1'+k_2,p} g_{k_2,p,p-k_1'+k_2}, \tag{A.3b}$$

$$D_{\text{ph2}}^{\vec{\eta}}(\vec{\xi}) = -2\sum_p \mathcal{L}_{p,p+k_1'-k_1}^{\text{ph}} g_{k_1,p+k_1'-k_1,k_1'} g_{k_1,p+k_1'-k_1,k_1'} g_{p,k_2,p+k_1'-k_1}, \tag{A.3c}$$

$$D_{\text{ph3}}^{\vec{\eta}}(\vec{\xi}) = \sum_p \mathcal{L}_{p,p+k_1'-k_1}^{\text{ph}} (g_{k_1,p+k_1'-k_1,p} g_{p,k_2,p+k_1'-k_1} + g_{k_1,p+k_1'-k_1,k_1'} g_{k_2,p,p+k_1'-k_1}). \tag{A.3d}$$

As explained in the main text, $g^{\vec{\eta}}(\vec{\xi})$ and $D_{\text{x}}^{\vec{\eta}}(\vec{\xi})$ on each side of (A.1) can be formally expanded in power of $\vec{\xi}$ to get the flow equations of the set of marginal and irrelevant couplings.

## A.2 Loop expressions

In order to derive the expressions of the bubble intensities, let us first introduce the free propagator regularized at scale $\Lambda$:

$$G_0^\Lambda(p_n, p) = \frac{\theta_\Lambda(p)}{\mathrm{i}p_n - \xi(p)}, \tag{A.4}$$

where $p_n$ denotes the fermionic Matsubara frequencies and $p$ the momentum. The loop expressions are then obtained from the derivative of the product of the propagators:

$$\mathcal{L}_{(p_n,p),(q_n,q)} = -\frac{\pi v_F T}{L} \Lambda \partial_\Lambda \big(G_0^\Lambda(p_n, p) G_0^\Lambda(q_n, q)\big), \tag{A.5}$$

and the sum over the Matsubara frequencies is then performed:

$$\begin{aligned}
\mathcal{L}_{p,q}^{\mathrm{pp}} &= \sum_{p_n} \mathcal{L}_{(p_n,p),(-p_n,-p+q)}, \\
\mathcal{L}_{p,q}^{\mathrm{ph}} &= \sum_{p_n} \mathcal{L}_{(p_n,p),(p_n,p+q)}.
\end{aligned} \tag{A.6}$$

The expressions of the loop contributions for the diagrams of the $\mathrm{p}-\mathrm{p}$ and $\mathrm{p}-\mathrm{h}$ scattering channels are thus given by

$$\begin{aligned}
\mathcal{L}_{p,q}^{\mathrm{pp}} &= -\frac{\pi v_F}{2L} \Lambda \partial_\Lambda\big(\theta_\Lambda(p)\theta_\Lambda(p+q)\big) \frac{n_F\big(\xi(p)\big) - n_F\big(-\xi(p+q)\big)}{\xi(p) + \xi(p+q)}, \\
\mathcal{L}_{p,q}^{\mathrm{ph}} &= \frac{\pi v_F}{2L} \Lambda \partial_\Lambda\big(\theta_\Lambda(p)\theta_\Lambda(p+q)\big) \frac{n_F\big(\xi(p)\big) - n_F\big(\xi(p+q)\big)}{\xi(p) - \xi(p+q)},
\end{aligned} \tag{A.7}$$

where $n_F(\xi) = (1 + \mathrm{e}^{\beta\xi})^{-1}$ is the Fermi-Dirac distribution and $\theta_\Lambda(k)$ is the regulator or cut-off function of the RG procedure. The latter is introduced explicitly in Sec. B below.

Let us discuss some limiting cases for these loops at vanishing external momentum. These enter in the flow equations of response functions. We can define the following intensities in each scattering channel. In the p-h channel, we have

$$\begin{aligned}
\mathcal{L}_{\mathrm{P}} &= \sum_{p\geqslant 0} \mathcal{L}_{p, p-2k_F}^{\mathrm{ph}} = \sum_{p\geqslant 0} \mathcal{L}_{-p, -p+2k_F}^{\mathrm{ph}}, \\
\mathcal{L}_{\mathrm{P}'} &= \sum_{p\geqslant 0} \mathcal{L}_{p, p+2k_F}^{\mathrm{ph}} = \sum_{p\geqslant 0} \mathcal{L}_{-p, -p-2k_F}^{\mathrm{ph}}, \\
\mathcal{L}_{\mathrm{L}} &= \sum_{p\geqslant 0} \mathcal{L}_{p, p}^{\mathrm{ph}} = \sum_{p\geqslant 0} \mathcal{L}_{-p, -p}^{\mathrm{ph}},
\end{aligned} \tag{A.8}$$

which correspond respectively to the $2k_F$ p-h or Peierls loops without ($\mathcal{L}_{\mathrm{P}}$) and with ($\mathcal{L}_{\mathrm{P}'}$) umklapp scattering, and to $q = 0$ p-h loop. As for the p-p or Cooper loop at zero pair momentum, it is given by

$$\mathcal{L}_{\mathrm{C}} = \sum_{p\geqslant 0} \mathcal{L}_{p, -p}^{\mathrm{pp}} = \sum_{p\geqslant 0} \mathcal{L}_{-p, p}^{\mathrm{pp}}. \tag{A.9}$$

These quantities are plotted in Fig. 19 as a function of the RG time $\ell$ defined by $\Lambda = \Lambda_0 \mathrm{e}^{-\ell}$. We can observe the presence of the van Hove singularity located at the edge of the spectrum. At half-filling the amplitudes of the Cooper and Peierls bubble intensities are the same at all $\ell$ but opposite in sign, and lead to maximum interference between certain classes of diagrams in Fig. 2. Away from half-filling, the Peierls intensity $\mathcal{L}_{\mathrm{P}'}$, which involves umklapp scattering, sees its intensity suppressed as a function of $\ell$, when typically $\Lambda(\ell) < v_F \mu$. This differs from

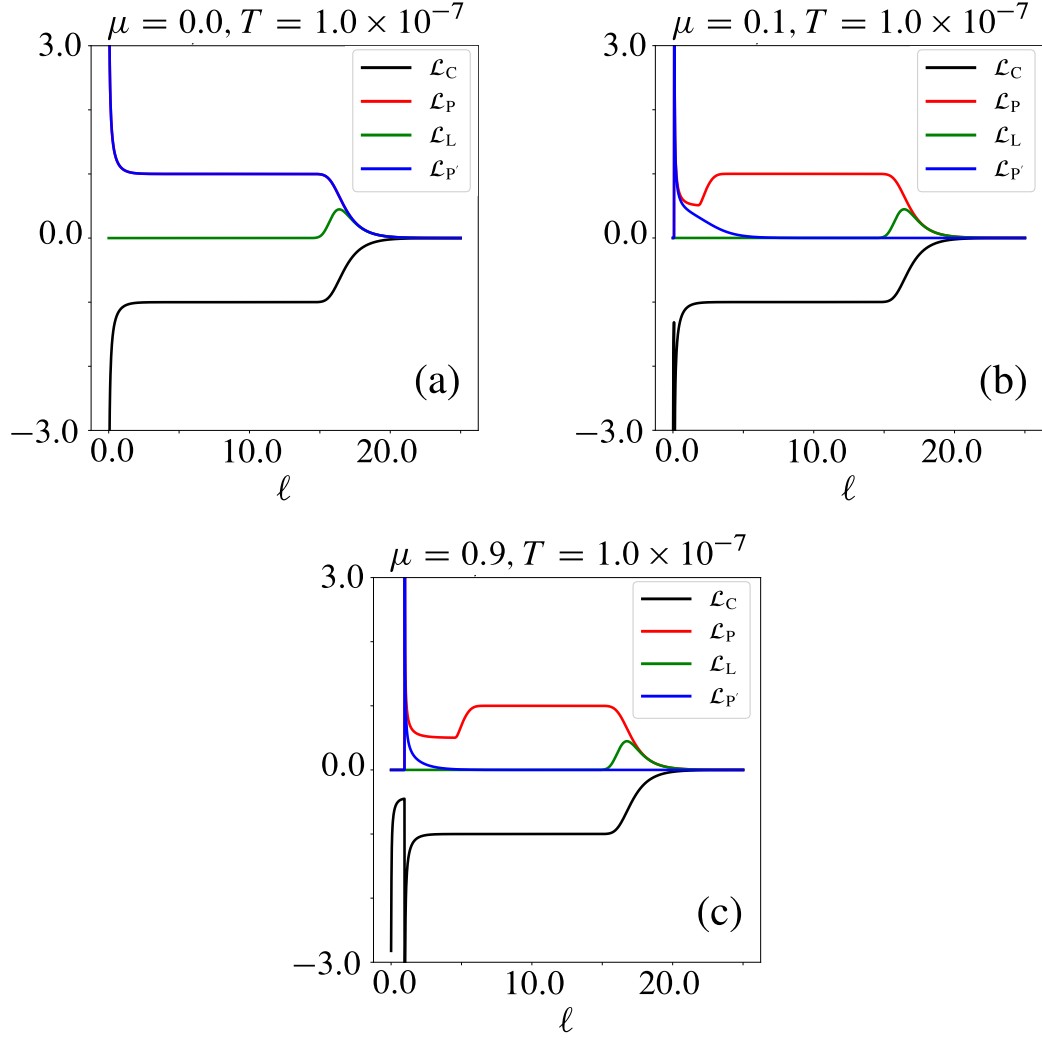

Figure 19: Cooper, Peierls and Landau bubbles shown for different values of the chemical potential, in the case of a tight-binding spectrum. The first panel is at half-filling and the others at different fillings. In this figure, $\ell$ is the RG time, defined by $\Lambda = \Lambda_0 e^{-\ell}$. The sharp peaks appear when $\Lambda$ hits the band edges and are due to the van Hove singularity in the density of states.

the normal part $\mathcal{L}_P$ with no umklapp scattering which keeps its full intensity down to the thermal shell. We also note in the third panel of Fig. 19 that at sizeable doping all the Peierls intensities are zero at the beginning of the flow. This results from the particle-hole asymmetry of the spectrum which suppresses electron or hole states required for $2k_F$ p-h pairing. By contrast the asymmetry of the spectrum suppresses only half of the states for p-p pairing states so that the Cooper intensity is only halved and remains finite at the beginning of the flow.

### A.3 Renormalization of the Fermi velocity

Let us compute the Hartree-Fock contributions to the renormalization of the one-body term. They can be put in the diagrammatic form

$$\partial_\Lambda \Sigma_k = \quad + \quad . \tag{A.10}$$

One can expand $\Sigma_k \simeq \Sigma_0 + \Sigma_1 \xi(k)$ to first order in the energy $\xi$. The momentum-independent term $\Sigma_0$ renormalizes the chemical potential. However, it can be rescaled back to its initial value (5) at each step of the flow for a given band-filling, so it is the bare value of $\mu$ that is used in the flow equations. The momentum-dependent term linear in $\xi$, $\Sigma_{1,\Lambda}$, leads to the flow of the hopping term $t_\Lambda$ or correspondingly of the renormalization of the Fermi velocity $v_{F\Lambda} = v_F(1 + \Sigma_{1,\Lambda})$.

From the evaluation of the second Fock term of (A.10), one has using a sharp cutoff

$$\partial_\Lambda v_{F\Lambda} = \frac{v_F^2}{8} \int_{-\Lambda_0}^{+\Lambda_0} d\xi \, \mathcal{N}(\xi) n_F\big(\xi(2 - v_{F\Lambda}/v_F)\big)$$
$$\times [\delta(\xi + \Lambda) + \delta(\xi - \Lambda)][g_{1,0,0}^{+,-,-} + (g_{1,0,1}^{+,-,-} + g_{1,0,1}^{+,+,+})\xi], \tag{A.11}$$

where the momentum-dependent backward and forward scattering amplitudes have been expanded up to second order in $\xi$ following the notation introduced previously in (42-43). In the low-temperature and low-energy limits, this equation becomes

$$\partial_\ell v_{F\ell} = \frac{1}{8} v_F\Big(\frac{V}{\pi v_F} e^{-2\ell} + \frac{V\mu}{\pi v_F} e^{-\ell}\Big). \tag{A.12}$$

This leads to the renormalized Fermi velocity in the low-energy limit

$$v_F^* = v_F\Big[1 + \frac{V}{8\pi v_F}\Big(\frac{1}{2} + \mu\Big)\Big]. \tag{A.13}$$

In weak coupling, $v_F^*$ differs from $v_F$ only by a few percents.

## B Choice of the regulator

The regulator $r_a(x)$ is realized as a smooth step function, and depends on a rigidity parameter $a$ (numerically $a \approx 10$), such that $r_{a=\infty}(x) = \Theta(x-1)$, where $\Theta(x)$ is the Heaviside function. Its expression is the following:

$$r_a(x) = g(ax - a + 1/2), \tag{B.1}$$

where

$$g(x) = \frac{f(x)}{f(x) + f(1-x)},$$
$$f(x) = \begin{cases} e^{-1/x} & \text{if } x > 0, \\ 0 & \text{otherwise} . \end{cases} \tag{B.2}$$

The regulator is shown in Fig. 20. It enters the flow equations through the function $\theta_\Lambda(k)$ in the regularized free propagator (see Eqs. (A.4)). This function only depends on the momentum $k$ through the variable $\xi(k) = \xi$, and is given by

$$\theta_\Lambda(k) = r_a(|\xi|/\Lambda). \tag{B.3}$$

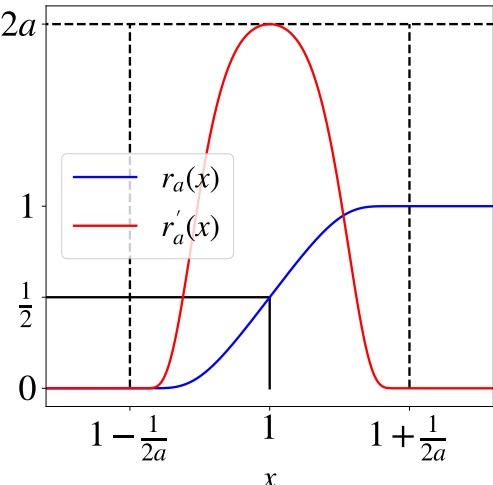

Figure 20: The regulator is such that $r_a(0) = 0$, $r_a(x \gg 1) = 1$ and $r_a(1) = 1/2$.

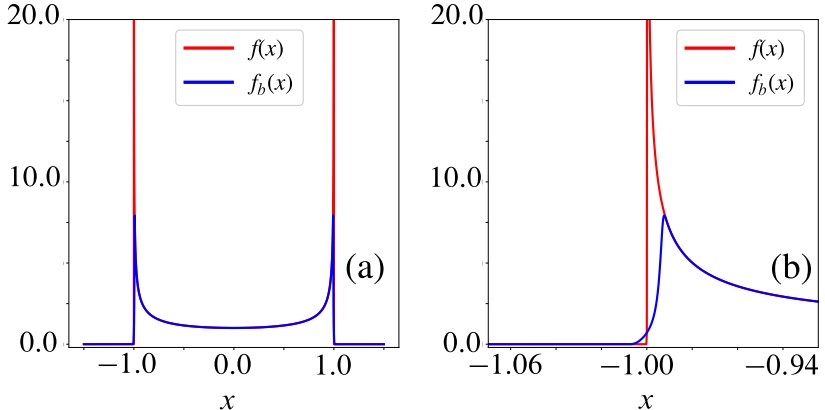

Figure 21: The van Hove singularity is regularized thanks to a smooth gate function $G_b(x)$, and the original density of states $f(x)$ is replaced by the regularized function $f_b(x)$ whose sharpness is controlled by the parameter $b$, with which we have $f_{b \to \infty}(x) = f(x)$.

Such a cutoff procedure is meant to reproduce the Wilsonian RG approach,[4] which amounts to a progressive integration of the degrees of freedom. Here, the UV degrees of freedom are integrated first, and the RG flow leads to a low-energy effective theory. In the case of one-dimensional fermions, the low-energy theory corresponds to a model with a linear spectrum comprising two branches centered around the two Fermi points. This is of course in stark contrast to the bosonic case for which the low-energy theory is described by modes of momenta $k \approx 0$.

Let us now clarify the structure of the bubble $\mathcal{L}_{p,q}^{\mathrm{ph,pp}}$. Each bubble is made of two factors: the first one is proportional to the cutoff function while the second is proportional to the derivative of this function with respect to the RG parameter $\Lambda$ — unslashed and slashed fermion lines respectively, in diagrams of Figs. 2, 3 and 4. Since the cutoff function is roughly a regularized step function, its derivative is a regularized Dirac function, whose effect is a selection of modes of energy $\xi \sim \Lambda$, and hence reproduces Wilson's idea.

---

[4]Indeed, one recovers a sharp cutoff in the limit $a \to \infty$, that is $r_\infty(|\xi|/\Lambda) = \Theta(|\xi| - \Lambda)$.

**Van Hove singularity regularization.** The regulator function $r_b(x)$ can be used to regularize the van Hove singularity. The density of states has the schematic form:

$$f(x) = \frac{\Theta(1-|x|)}{\sqrt{1-x^2}}, \tag{B.4}$$

and is singular at $x = \pm 1$. In order to regularize this function, we first define a regularized gate function:

$$G_b(x) = r_b(x+2)\big(1 - r_b(x)\big), \tag{B.5}$$

and then make the following replacement:

$$f(x) \to f_b(x) = \frac{G_b(x)}{\sqrt{1 - x^2 G_b(x)}}. \tag{B.6}$$

The regularized van Hove singularity is shown in Fig. 21. Such a regularization is advantageous, because it produces a smooth function, well suited for numerical evaluations. Furthermore, the error due to the regularization is restricted to small segments around the singular points. This is because the regulator is built out of functions whose variation support is compact. The total number of states is recovered in the limit $b \to \infty$ (numerically $b \approx 10^3$).

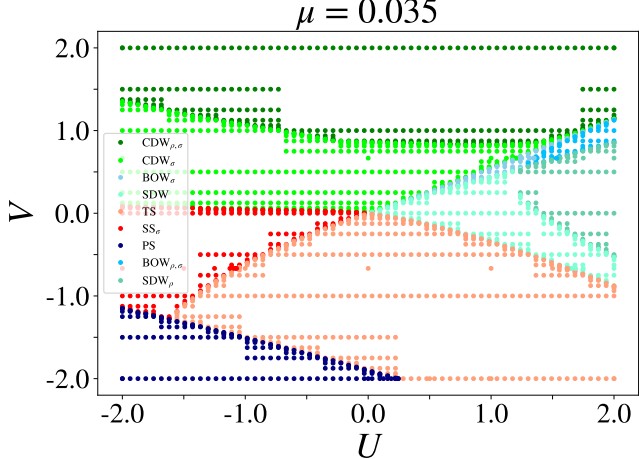

Figure 22: An example of a raw phase diagram obtained by the dichotomy algorithm explained in the text and which leads to Fig. 9 with continuous boundaries.

# C  Numerical determination of phase boundaries

Boundaries of the phase diagrams are determined using a dichotomy algorithm. The algorithm is initialized by specifying several parameters:

- $V_- < 0$ and $V_+ > 0$, with typically $V_+ = -V_- = 2$,

- $U_- < 0$ and $U_+ > 0$, with typical values $U_+ = -U_- = 2$,

- a number $N$ which determines vertical lines $U_i = U_- + i(U_+ - U_-)/N$ for $0 \leqslant i \leqslant N$ ($N \approx 60$),

- a number $G > 1$ specifying the total number of dichotomy iterations (in practice $G = 10$).

In the first step, the phase is determined on each point of coordinates $(U_i, V_\pm)$. Then a dichotomy is performed on each vertical line, until the final number of generations is reached. The vertical line setting is convenient because the computations done on two different lines are independent from each other, which allows the use of parallelization. An example of a raw phase diagram is shown in Fig. 22.

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
