# Peer review of "Functional Renormalization Group for fermions on a one dimensional lattice at arbitrary filling"

_SciPost Physics, doi:SciPost Phys. 17, 054 (2024)_

## Round 2 · Referee Report · Daniel Rohe (Referee 1) · 2023-11-6

Strengths

  1. The work presents interesting results on so far not or little explored regions of the phase diagram, in particular on but not restricted to the aspect of a bond-order wave that appears "on top" of the transition region defined by U=2V between charge-density and spin-density waves for U,V>0. The evolution of this peculiar behaviour when moving away from half filling is outlined and discussed in detail, along with several other aspects. The underlying data on which the deduction of the phase diagram is based is discussed and presented in detail.

  2. The manuscript is well structured and gives a systematic presentation of model, method and results.

  3. Results are validated against previous/exisiting works in detail, before new results are presented.

Weaknesses

  1. At some stages I felt some more information could be included on the formal level as well as concerning auxiliary parameters.

  2. Some physical aspects may deserve to be outlined with a little more clarity. I tried to make some suggestions on these matters in the comments below.

Report

The manuscript "Functional renormalization group for fermions on a one-dimensional lattice at arbitrary filling" presents a new application of functional renormalization group methods to the extended Hubbard model in one dimension. In particular, the authors consider the non-linearity of the dispersion as well as the momentum dependence of the flowing couplings. The scheme is first outlined and developed, and then applied numerically. It is found to reproduce previous results at half filling for the phase diagram and is then applied to non-half-filled (i.e. "doped") cases, for which it uncovers several new aspects and observations.

In summary, I consider the manuscript to be suitable for publication in SciPost Physics. I do offer some remarks, questions and suggestions below, the main intention being to provide constructive input.

Requested changes

My comments below are above all meant as questions, suggestions and remarks, rather than firmly requested changes. In case I might ask for information that is actually given in the manuscript, please ignore the respective comment(s). I cannot rule out that I may have overlooked something.

General comment:

i) It is somewhat implied in abstract and introduction, that the matter of interest are ground state properties. This is however not made very explicit, and the numerical calculations are then actually done at a small but finite temperature, if I understand correctly. It seems appropriate to me to outline and justify this procedure more explicitly, also in light of the generically delicate situation of long-range order - or rather its absence - in 1d (quantum) systems even at T=0.

a) I would find it helpful to clarify this aspect early in the manuscript and to state the actual temperature value that was chosen for the numerics. While a value T=10^(-7) is stated once in the caption of Figure 10, it is however not clear to me if this is the general value chosen for all numerical calculations.

b) In the conclusion it is stated that "the nature of ground states" was checked. A comment in how far and for which type of quantities/observables a small but finite temperature in this numerical approach allows conclusions about the ground state would be beneficial to corroborate the conclusions.

c) Technically, (1-PI) fRG computations can be done at T=0, as employed in previous works, sometimes even being a preferred choice. What is the reason for not doing this here? Are there singular contributions at T=0 that are not regularised by the chosen momentum cut-off? -> A brief sentence stating the reason for the actual value of T that was chosen in contrast to T=0 seems helpful to me.

Section 2:

ii) Equation (6): The quantity 'L' could/should be defined here already.

iii) Equation (13): Extracting a factor 'T' from the coupling function is unfamiliar to me. In particular, it makes the limit T->0 appear awkward. Is there a necessity or deeper reason to do this? It also seems to collide with equation (8), if I'm not mistaken.

iv) Equation (14) - concerning the regulator:

a) I would already at this stage briefly but explicitly state the choice of the actual regulator that is used, or at least point to the appendix.

b) Eq. (23) suggest a sharp cut-off by virtue of Theta functions, while in appendix A.3 it is outlined in detail that it is actually a smooth cut-off. Adding the parameter 'a' to these Theta functions in the equations/definition of the main body could avoid this possible misinterpretation.

c) For completeness, the chosen value of the "smoothness" parameter 'a' could/should be explicitly specified and be related to other quantities with which it 'numerically interferes', such as the temperature and the lowest cut-off value that is reached in the computations.

v) Related to this, in Figures 2,3: The dashed/"derived" propagators are somewhat loosely defined in the caption as "line in the outer shell". In 1-PI fRG they are more generally defined as "single-scale" propagators (e.g. Ref. 30) and it may be worthwhile to define them as such briefly but more precisely in the main body, in particular since the scheme employs a slightly softened cut-off.
Overall, a more explicit statement on the elements that are depicted in the two figures would improve clarity, it need not be long.

vi) Equation (20): Again, the factor 'T' puzzles me, c.f. comment iii).

vii) Section 2.3: The paragraph between equation (20) and (21) is somewhat unclear to me. Formally, the 1-PI fRG equations are exact, with a regulator being implemented in the quadratic part of the bare action (only). The coupling function, in turn, is always defined everywhere for all momenta and (usually) not subject to a separate, additional cut-off. Also, since the non-derived propagators (for a momentum cut-off) live above the cut-off energy, I would not expect "unavailable states" in the low-energy section of the flow, in contrast to the reasoning provided in the text by "...namely above the scaled energy \Lambda of integrated degrees of freedom".
It is unclear to me if such a function is also used in the numerics or only required to make contact with the g-ology continuum model.
Maybe this part can be made clearer, potentially also by adding a reference. It could however also be that this question is due to my personal (lack of) understanding, and that I am simply not familiar with this type of procedure. My feeling is that this modification might owe to procedures that are common in other types of g-ology RG treatments.

viii) Equation (23): Here, it is implicitly suggested that self-energy effects are neglected, since bare propagators are used. Yet, it is later stated that "some" self-energy corrections are actually implemented by means of a renormalised Fermi velocity, below Eq. (33). Thus, the propagators inside the loops are not really bare G^0 entities, if I understand correctly. This could be outlined earlier, c.f. comment v), also since in 1-PI fRG the loop contributions when going beyond G^0 and including a flowing self-energy (not done here) cannot generally be written as \Lambda-derivatives of bubbles - c.f. comment v) about single-scale propagators.

ix) Section 2.3, second last sentence before section 2.3.1 - "... we do not consider uniform q->0 responses": I assume this is essentially due to the fact that potentially relevant forward scattering contributions in a scheme with a (nearly) sharp momentum cut-off only enter the flow asymptotically for \Lambda->0, and thus cannot properly compete with the contributions of interest here, which begin to flow to strong coupling at higher values of the cut-off. That of course is a subtle matter and may also depend on the ratios of the different "smoothness" parameters that are used, see also comment i)a) and iv)c).
-> Maybe it could be mentioned which kinds of ordering tendencies are thereby a priori "deselected" and in how far this constitutes a restriction - or not - on what follows. Are there parameter regions in the phase diagram where q->0 responses could play a relevant role, like it is the case e.g. for ferromagnetism in 2d?

x) Section 2.3.1: The bubble "intensities" are stated. I would find it useful to include an explicit computation, which should be rather concise but could be helpful. It could be included/added in Appendix A.2.

xi) Section 2.3.2: The first sentence mentions the "low-temperature" limit. Can this be better quantified/specified, c.f. comment i)? Is it generally the value given in the caption of Figure 10, i.e. 10^(-7)?

xii) Fig. 5 lacks the indications '(a)' and '(b)' - it is obvious what is meant, though.

xiii) A general remark on the graphical presentation of the phase diagrams: They appear as continuous diagrams with sharply defined and continuous transition/separation lines. I assume that in practice numerous fRG runs have been conducted for a number of parameter sets to map this out. Maybe it is possible to indicate this somehow, at least as an example in one of the diagrams, to be able to relate the continuously depicted diagrams to the actual set of results from which they are deduced, similar to e.g. Fig. 8 in DOI 10.1103/PhysRevB.61.7364 .

xiv): Section 2.4 - concerning the self-energy corrections: They are included in terms of a renormalised Fermi velocity v_F which is calculated "in the scaling limit".

a) How is "scaling limit" meant here? In the scope of 1-PI fRG there is no rescaling involved in the formalism.

b) Would it be feasible (and worthwhile at all) to include v_F as a flowing quantity, and at what cost and effort? C.f. comment viii).

c) Why is it better or maybe even necessary to work with the "final" v_F rather than the initial/bare value? Does this lead to relevant qualitative/quantitative changes in the results?

xv) On the general strategy to expand the effective interaction in \xi: Would it also be possible/feasible to work with a more direct and sufficiently fine "brute-force" patching of the effective interaction in momentum space, such as e.g. in various other (2d) fRG works? This would of course increase the number of couplings constants that have to be treated numerically, but in light of 2d calculations based on that technique I would (naively) expect this to be feasible.

Section 3:

xvi) First paragraph - "The calculations are limited to the weak-coupling sector": It might be worth briefly stating the reason, assumably the truncation of the 1-PI fRG hierarchy. This could also be mentioned earlier in the text, c.f. comments iv) and v).

xvii) Figure 7,8:

a) The y-axes could be labelled explicitly in the plots.

b) Explicit quantitative information on the actual value of the low-energy scale at which the flow is stopped would be nice, and how this relates to the temperature that is chosen for the numerics, to better understand the mutual relevance of the various low-energy scales - c.f. comment i).

c) Fig. 8 is discussed in the text before Fig. 7. Both figures show results for specific points in Fig. 6.
-> Merging the two Figures into one might be an option for better readability and a more direct view of the underlying results.

xviii) Figure 11: I find the legends to be a little small.

xix) Section 3.2 - third paragraph - "... we can write N(\xi)=1/(Pi*v_F)": Since v_F is renormalised, c.f. comment xiv), shouldn't this be the renormalised value? This would then differ somewhat from g-ology, wouldn't it?

xx) Section 3.2.1 - last paragraph: "... in the second quadrant": It was not clear to me which one is the "second". I'd suggest to use "lower left", "lower right", etc., to avoid ambiguities. Or it might be an option to add thin lines to separate the quadrants and to label them explicitly as e.g. I, II, III and IV. The mutually distinct nature of the physics would justify that, in my opinion.

xxi) A general question and a mere matter of interest: With increasing temperature I would expect the flow to become regularised at some stage, and this should define a "one-loop T_c", i.e. a physical "cross-over/binding/short-range-ordering" temperature. Has this been looked at in this context?

xxii) Section 3.2.3 - page 27 lower part: "Calculations carried out ... namely up to \mu=sqrt(2) (3/4 filling)": Why is this value chosen as the upper limit for the calculations? What would happen beyond? Is the method still applicable then? If not why not? Or would it simply go beyond the scope of this work? C.f. comment xxiii) -> quarter filling.

Section 4:

xxiii) It is mentioned that various previous results could be confirmed. The new results are then presented stating that "We have also carried out ... away from half filling": To my personal taste it would well be worth emphasising a bit more that these are new results, by which unexplored terrain is being entered. These are in my view the most striking results contained in the manuscript, drawing their solidity of course from the fact that the method is in line with previous results.
That said, there is some previous work on the special case of quarter filling, e.g. DOI 10.1103/PhysRevB.75.113103, and likely/possibly others (I did not manage to do a comprehensive research on this). If possible and sensible, it might be useful to compare against such prior results, too. It would be interesting if and how quarter filling might emerge as a special case within the formalism presented here. That can of course also be left for future work.

xxiv) Some words on the specifics of the numerical implementation and the computational costs would be of interest, also to have an idea of what is possible and feasible in such a set-up. Ideally, the underlying code might be worth being developed further and even be published as a result in its own right.

xxv) As part of the outlook, anisotropic systems are mentioned. These systems may actually permit to compare the extension of typical 1d (f)RG schemes to extensions of typical 2d (f)RG schemes, the two approaches often being quite different in nature. I wonder how the authors would feel about this aspect.

xxv) Appendix A.3, c.f. also comment iv): There are two parameters 'a', but for two different purposes. Also, the actual values that are used in the numerics are not specified (or I did not find them). I suggest to explicitly name them as two separate parameters and to provide the numerical values that were used.

---

## Round 2 · Referee Report · Anonymous (Referee 2) · 2023-11-10

Strengths

  1. The paper extends the fRG analysis to away-from-half-filling conditions in 1D lattice models for interacting fermions, addressing the lack of particle-hole symmetry and its consequences, which is a valuable extension of the research.

  2. This fRG approach allows for systematic consideration of lattice effects on the effective continuum field theory at low energies.

  3. The detailed explanation of how the flow of coupling constants is conducted in each case is very informative and beneficial.

Weaknesses

  1. The paper could improve by delineating its advancements over existing fRG methods more explicitly.

  2. The approach presented in this paper may be highly technical and complex, which could limit its accessibility to readers who are not specialists in the field, although this is somewhat unavoidable.

Report

This manuscript introduces a weak-coupling functional renormalization group (fRG) approach for one-dimensional (1D) interacting fermion lattice models, applicable away from half filling, using the extended Hubbard Model (EHM) to show how lattice effects influence low-energy effective field theories. It finds that at half filling, irrelevant interactions affect phase transitions, with a notable impact on BOW, SDW, and superconducting states. Away from half filling, the loss of particle-hole symmetry alters high-energy state contributions to phase stability and transitions, leading to deviations from traditional continuum theory predictions. The findings of this paper align with previous research and suggest broader applicability to various lattice models, with potential extensions to explore doped fermionic systems.

Applying the fRG approach to cases away from commensurate fillings in interacting electron models is challenging, yet this paper presents one such method and meticulously explains its effectiveness when applied to the EHM. This provides valuable information for the future development of this technique. Therefore, if the manuscript is sufficiently improved by addressing the questions/suggestions I have listed below as much as possible, I would recommend its publication in SciPost Physics.

(1) The authors state in the introduction that 'These RG results were strictly speaking limited to the EHM model at half-filling.' Does this mean that traditional RG methods were not applicable when deviating from half-filling, or that they could be applied but did not yield good results? Clarifying these points and further highlighting the technical contributions developed in this paper would enhance the understanding of its significance.
(2) This may be a trivial question on Fig.1: Why is the density of states not symmetric with respect to the band center (\epsilon=0) for the tight-binding model? Is there already some form of interaction incorporated in a circular manner?
(3) It is unclear if the flow equations Eqs.(49-51) for channels at q=\pm2k_F are still applicable to the case away from half filling.
(4) While the qualitative agreement with previous works at half-filling is acknowledged, it is not clear if the results at half filling are also improved by the presented fRG approach?
(5) At small doping levels, how can it be determined that the charge gapless CDW, BOW, and SDW phases are incommensurate? It is presumed that charge and/or spin gapped phases are commensurate, but can commensurate CDW or SDW phases still occur when the system deviates from half-filling? Moreover, while the concept of incommensurate CDW and SDW is comprehensible, what constitutes an incommensurate BOW state because BOW state is characterized by a two-site unit cell?
(6) Related to question (5), when investigating the EHM lattice model using numerical methods such as DMRG or QMC, it is anticipated that away from half-filling, the charge gap in the repulsive region would close immediately. To put it plainly, at V=0, the Bethe Ansatz indicates that only at n=1 is there a charge gap due to a singularity, but it is evident that deviating from n=1 leads to a charge gapless state. However, Fig. 9 still shows a charge gapped state for U>1.5. How can this discrepancy be understood?
(7) It would be helpful to the general reader if the paper could provide a clearer explanation of how the gapful and gapless states in the charge and spin sectors are determined.
(8) Some figures on flow of the vertices and couplings are missing titles on the vertical axis. It would be beneficial for clarity and completeness if these could be added.
(9) The boundary between the SDW and TS phases is indicated to be in the second quadrant, while the boundary between the CDW and SS phases is stated to be in the fourth quadrant. Could there be a possibility that this is opposite?
(10) The 1D doped EHM has been studied by an fRG method in a relatively recent paper: Y.-Y. Xiang et al., J. Phys.: Condens. Matter 31 (2019). A brief statement about discrepancy to this paper may be useful.
(11) Please correct some typos, e.g., abbreviation `EHM' is twice defined, `TL' is defined in the second appearance of Tomonaga-Luttinger, there is no vector notation (arrow) in |n|\neq0, etc.

---

## Round 4 · Referee Report · Daniel Rohe (Referee 1) · 2024-6-13

Report

The authors provide a thorough revision of the original submission. All points suggested from my side in the first report phase have been properly addressed. Beyond this, the content has actually been extended. This exceeds my intention and expectations, and I sincerely acknowledge the effort that the authors have invested in this major update.

I only have two remaining/additional minor comments, both fully optional:

i) With eq (23) and (24) having been added, I experienced a short struggle concerning indices. 's' is double-used as an upper index, obviously either as "singlett" or "site". Also 'b' is double-used, as an upper index for "bond", and as a lower index for summation. I had to look twice to resolve this. If it is possible to increase clarity in that area, it might improve the readability.

ii) Please check if the following suggestion might be favourable or not:

On page 11 third paragraph, the conditions "U > \pm 2V" and "V < \mp U/2" might be expressed more precisely as "U > |2V|" and "V < - |U/2|"

Requested changes

None. C.f. optional suggestions above.

Recommendation

Publish (easily meets expectations and criteria for this Journal; among top 50%)

---

## Round 4 · Referee Report · Anonymous (Referee 2) · 2024-6-23

Report

The paper appears to have been significantly improved by incorporating the comments from both referees. The clarity has greatly improved, making it very readable and understandable even for non-experts. Therefore, I recommend the publication of this paper in SciPost Physics in its current form.

Recommendation

Publish (easily meets expectations and criteria for this Journal; among top 50%)

---

## Round 4 · Author Response

Dear Editor,

\medskip

we would like to thank the referees for their stimulating questions and comments. We have tried to address them as fully as possible and hope that this has helped to improve the manuscript. We apologize for the long delay before resubmission, which is largely due to the many changes we have made to the manuscript.

\medskip

Sincerely Yours,

\medskip

Lucas Désoppi, Claude Bourbonnais and Nicolas Dupuis

---

## Round 4 · List of Changes

\newcommand{\comment}[1]{\textcolor{blue}{\textbf{Authors' comments:} #1}}
\newcommand{\modif}[1]{\textcolor{blue}{\textbf{Modifications to the manuscript:} #1}}

\section*{Reply to the first referee}

i) It is somewhat implied in abstract and introduction, that the matter of interest are ground state properties. This is however not made very explicit, and the numerical calculations are then actually done at a small but finite temperature, if I understand correctly. It seems appropriate to me to outline and justify this procedure more explicitly, also in light of the generically delicate situation of long-range order – or rather its absence – in 1d (quantum) systems even at $T=0$.

a) I would find it helpful to clarify this aspect early in the manuscript and to state the actual temperature value that was chosen for the numerics. While a value $T=10^{-7}$ is stated once in the caption of Figure 10, it is however not clear to me if this is the general value chosen for all numerical calculations.

b) In the conclusion it is stated that "the nature of ground states" was checked. A comment in how far and for which type of quantities/observables a small but finite temperature in this numerical approach allows conclusions about the ground state would be beneficial to corroborate the conclusions.

c) Technically, (1-PI) fRG computations can be done at $T=0$, as employed in previous works, sometimes even being a preferred choice. What is the reason for not doing this here? Are there singular contributions at $T=0$ that are not regularised by the chosen momentum cut-off? $\to$ A brief sentence stating the reason for the actual value of $T$ that was chosen in contrast to $T=0$ seems helpful to me.

\medskip

\comment{The temperature is chosen extremely small so that all the phase diagrams remain rigorously the same when the temperature is further reduced. When a gap opens for instance, it is larger than the typical RG scale associated to temperature. We choose to work at an arbitrary - very small - temperature so that Fermi-Dirac distributions remain smooth (instead of dealing with $T=0$ step functions), which makes it easier to deal with in numerical computations.}

\medskip

\modif{A general remark concerning the temperature at which calculations are done is made at the beginning of the Section 3 on lattice results on page 17. }

\bigskip

ii) Equation (6): The quantity $L$ could/should be defined here already.

\medskip

\modif{The number of lattice sites $L$ has been defined just before equation (6).}

\bigskip

iii) Equation (13): Extracting a factor $T$ from the coupling function is unfamiliar to me. In particular, it makes the limit $T \rightarrow 0$ appear awkward. Is there a necessity or deeper reason to do this? It also seems to collide with equation (8), if I'm not mistaken.

\medskip

\comment{True, in the absence of any summations the $\pi v_F T/L$ factor was misplaced in (13) and should appear elsewhere. }

\medskip

\modif{The factor has been moved in front of the interaction term in (10) where all the summations now explicitly appear.}

\bigskip

iv) Equation (14) - concerning the regulator:

a) I would already at this stage briefly but explicitly state the choice of the actual regulator that is used, or at least point to the appendix.

b) Eq. (23) suggest a sharp cut-off by virtue of Theta functions, while in appendix A.3 it is outlined in detail that it is actually a smooth cut-off. Adding the parameter $a$ to these Theta functions in the equations/definition of the main body could avoid this possible misinterpretation.

c) For completeness, the chosen value of the "smoothness" parameter $a$ could/should be explicitly specified and be related to other quantities with which it 'numerically interferes', such as the temperature and the lowest cut-off value that is reached in the computations.

\medskip

\comment{b) The regulator appearing in this section is sharp, which is convenient to push the analytical computations one step further. However, in the rest of the text, it is the regulator as defined in the appendix. c) The parameter $a$ is introduced just for mathematical reasons. }

\medskip

\modif{a) The reference to the appendix has been added. b-c) It is now written that the regulator appearing in this section is sharp, which is a limiting case of the expression (71) of the Appendix B. This allows a connection with the known results of the continuum limit of the electron gas model. }

\bigskip

v) Related to this, in Figures 2,3: The dashed/"derived" propagators are somewhat loosely defined in the caption as "line in the outer shell". In 1-PI fRG they are more generally defined as "single-scale" propagators (e.g. Ref. 30) and it may be worthwhile to define them as such briefly but more precisely in the main body, in particular since the scheme employs a slightly softened cut-off.
Overall, a more explicit statement on the elements that are depicted in the two figures would improve clarity, it need not be long.

\medskip

\modif{The standard definition is now given in the text, and has been used in the caption of Fig. 2.}

\bigskip

vi) Equation (20): Again, the factor $T$ puzzles me, c.f. comment iii).

\medskip

\comment{This ambiguity has been dispelled in addressing point iii) above.}

\bigskip

vii) Section 2.3: The paragraph between equation (20) and (21) is somewhat unclear to me. Formally, the 1-PI fRG equations are exact, with a regulator being implemented in the quadratic part of the bare action (only). The coupling function, in turn, is always defined everywhere for all momenta and (usually) not subject to a separate, additional cut-off. Also, since the non-derived propagators (for a momentum cut-off) live above the cut-off energy, I would not expect "unavailable states" in the low-energy section of the flow, in contrast to the reasoning provided in the text by "...namely above the scaled energy $\Lambda$ of integrated degrees of freedom".
It is unclear to me if such a function is also used in the numerics or only required to make contact with the g-ology continuum model.
Maybe this part can be made clearer, potentially also by adding a reference. It could however also be that this question is due to my personal (lack of) understanding, and that I am simply not familiar with this type of procedure. My feeling is that this modification might owe to procedures that are common in other types of g-ology RG treatments.

\medskip

\comment{This cutoff function indeed does not appear naturally in 1PI fRG approach. This function would in turn be natural in the Wilsonian RG approach to the renormalization of the microscopic action in which momentum summations are progressively limited by the scaled cutoff. The 1PI formalism is such that away from half-filling, the umklapp processes are progressively decoupled from the normal processes, in accordance with the Wilsonian approach. We carefully checked that the presence of this function does not bring any change to all phase diagrams at both qualitative and quantitative levels, so we decided to remove it from our discussion. }

\medskip

\modif{References to this cutoff function have been suppressed in the text and in the equations.}

\bigskip

viii) Equation (23): Here, it is implicitly suggested that self-energy effects are neglected, since bare propagators are used. Yet, it is later stated that "some" self-energy corrections are actually implemented by means of a renormalised Fermi velocity, below Eq. (33). Thus, the propagators inside the loops are not really bare $G^0$ entities, if I understand correctly. This could be outlined earlier, c.f. comment v), also since in 1-PI fRG the loop contributions when going beyond $G^0$ and including a flowing self-energy (not done here) cannot generally be written as $\Lambda$-derivatives of bubbles - c.f. comment v) about single-scale propagators.

\medskip

\comment{Since the corrections to the Fermi velocity are relatively small (few percents) in weak coupling, they essentially don't affect the results so that only bare propagators have been used in all flow equations. This is now explicitly mentioned in the main text. The derivation of the flow equation for the Fermi velocity now appears as a complement of information in Appendix (Sec.~A.3). }

\bigskip

ix) Section 2.3, second last sentence before section 2.3.1 - "... we do not consider uniform $q \to 0$ responses": I assume this is essentially due to the fact that potentially relevant forward scattering contributions in a scheme with a (nearly) sharp momentum cut-off only enter the flow asymptotically for $\Lambda\rightarrow 0$, and thus cannot properly compete with the contributions of interest here, which begin to flow to strong coupling at higher values of the cut-off. That of course is a subtle matter and may also depend on the ratios of the different "smoothness" parameters that are used, see also comment i)a) and iv)c).
$\to$ Maybe it could be mentioned which kinds of ordering tendencies are thereby a priori "deselected" and in how far this constitutes a restriction – or not – on what follows. Are there parameter regions in the phase diagram where $q \to 0$ responses could play a relevant role, like it is the case e.g. for ferromagnetism in 2d?

\medskip

\comment{
There are essentially two possibilities of uniform density ordering at zero temperature: phase separation and ferromagnetism. These result from a $q\to0$ singularity in the charge compressibility and spin susceptibility, respectively. As shown by numerical exact diagonalizations on the EHM studies at half-, quarter- and 2/3-filling [48,49], phase separation is well known to dominate over all other phases - including triplet and singlet superconductivity - in a finite region of the phase diagram at $V<0$. This region actually encroaches on the weak coupling domain considered in our work. Ferromagnetism, however, is not seen numerically at the above selected fillings, but is likely to show up in the limit of very small band filling (or close to complete filling) in the repulsive part of the $(U,V)$ plane, namely when the density of states becomes sufficiently magnified by the proximity of the van Hove singularity at the band edge. In 2D the influence of van Hove singularity appears at half-filling and ferromagnetism can occur when other instabilities are sufficiently weakened by next to nearest-neighbor hoppings [C. Honerkamp and M. Salmhofer PRB {\bf 64}, 184516 (2001).]
\\
\indent{}In the framework of the fRG, the uniform charge and spin susceptibities can easily be computed, using the fact that the degrees of freedom contributing to the p-p and $2k_F$ p-h fluctuations, corresponding to energies $|\xi|\gtrsim T$, are separated from those contributing to the $q\to 0$ response functions, corresponding to $|\xi|\lesssim T$. In a first step, the flow equations are integrated with $\Lambda$ running between $\Lambda_0$ and $T$, considering only the logarithmically-singular Cooper and Peierls channels and ignoring the $q=0$ (Landau) channel. The renormalized coupling constants at the energy scale $T$ are then used to compute the uniform susceptibilities. Instead of integrating the flow with $\Lambda$ running between $T$ and 0, one can simply use an RPA, which is known to be exact for a linear spectrum (and equivalent to bosonization) once fluctuations due to back scattering and umklapp processes have been integrated out. For a sufficiently small temperature, the results effectively correspond to the $T=0$ limit.}

\medskip

\modif{In view of the possibility that uniform ordering may affect the phase diagram of the EHM, we decided to include the flow equations of uniform compressibility and spin susceptibility at the one-loop RPA level. These now appear at the end of Section 2 of the manuscript. The region of stability for phase separation in the phase diagrams is shown and discussed for all fillings considered when lattice effects are included. }

\bigskip

x) Section 2.3.1: The bubble "intensities" are stated. I would find it useful to include an explicit computation, which should be rather concise but could be helpful. It could be included/added in Appendix A.2.

\medskip

\modif{A paragraph of section A.2 of the Appendix A has been added to provide a short derivation of the bubble intensities.}

\bigskip

xi) Section 2.3.2: The first sentence mentions the "low-temperature" limit. Can this be better quantified/specified, c.f. comment i)? Is it generally the value given in the caption of Figure 10, i.e. $10^{-7}$?

\medskip

\comment{This point has been addressed above (See i).}

\bigskip

xii) Fig. 5 lacks the indications '(a)' and '(b)' - it is obvious what is meant, though.
Ok.

\medskip

\modif{Indications (a) and (b) have been added.}

\bigskip

xiii) A general remark on the graphical presentation of the phase diagrams: They appear as continuous diagrams with sharply defined and continuous transition/separation lines. I assume that in practice numerous fRG runs have been conducted for a number of parameter sets to map this out. Maybe it is possible to indicate this somehow, at least as an example in one of the diagrams, to be able to relate the continuously depicted diagrams to the actual set of results from which they are deduced, similar to e.g. Fig. 8 in DOI 10.1103/PhysRevB.61.7364 .

\medskip

\modif{The section C has been added in the appendix in which the algorithm used to determine the phase boundaries from a "raw" diagram is explained.}

\bigskip

xiv): Section 2.4 - concerning the self-energy corrections: They are included in terms of a renormalised Fermi velocity $v_F$ which is calculated "in the scaling limit".

a) How is "scaling limit" meant here? In the scope of 1-PI fRG there is no rescaling involved in the formalism.

b) Would it be feasible (and worthwhile at all) to include $v_F$ as a flowing quantity, and at what cost and effort? C.f. comment viii).

c) Why is it better or maybe even necessary to work with the "final" $v_F$ rather than the initial/bare value? Does this lead to relevant qualitative/quantitative changes in the results?

\medskip

\comment{a) The term "scaling limit" may appear indeed a bit unclear. It simply means a large $\ell$ regime where all transients and irrelevant terms do not contribute anymore. b-c) The inclusion of a scale-dependent (i.e. $\ell$-dependent) Fermi velocity is feasible, but would require a significant amount of modifications to the existing program. In weak coupling these effects can be ignored, so that we consider only the initial value of the Fermi velocity. See also viii) above.}

\medskip

\modif{The expression "scaling limit" in text has been replaced by "low-energy limit".}

\bigskip

xv) On the general strategy to expand the effective interaction in $\xi$: Would it also be possible/feasible to work with a more direct and sufficiently fine "brute-force" patching of the effective interaction in momentum space, such as e.g. in various other (2d) fRG works? This would of course increase the number of couplings constants that have to be treated numerically, but in light of 2d calculations based on that technique I would (naively) expect this to be feasible.

\medskip

\comment{It is indeed possible to work with a patch approach. It has been done for one-dimensional extended Fermi-Hubbard model at half-filling (see e.g. reference [28]). The patch procedure is less transparent, however, in the sense that all marginal and irrelevant coupling terms are embedded in the momentum - patch - dependence of interaction. Our approach allows to recover the standard g-ology flow equations in the low energy limit; in general its numerical cost is significantly less than the patch approach. }

\bigskip

xvi) First paragraph - "The calculations are limited to the weak-coupling sector": It might be worth briefly stating the reason, assumably the truncation of the 1-PI fRG hierarchy. This could also be mentioned earlier in the text, c.f. comments iv) and v).

\medskip

\comment{It is indeed because the infinite hierarchy of flow equations has to be truncated, and in the case of fermions, means that one has to restrict ourselves to the weak coupling limit.}

\medskip

\modif{This precision has been incorporated earlier in the main text, at the end of Sec. 2.2.}

\bigskip

xvii) Figure 7,8:

a) The y-axes could be labelled explicitly in the plots.

\medskip

\modif{The y axis label has been added.}

\bigskip

b) Explicit quantitative information on the actual value of the low-energy scale at which the flow is stopped would be nice, and how this relates to the temperature that is chosen for the numerics, to better understand the mutual relevance of the various low-energy scales - c.f. comment i).

\medskip

\comment{The flow is stopped when one of the following conditions is fulfilled: i) there is a divergence, signalling strong coupling and the opening of a gap; ii) in the absence of any gaps the RG flow can be integrated up to $\ell \to \infty$ where it is automatically cut off by the Fermi distribution factor at an arbitrary chosen low temperature (here $T=10^{-7}$). It follows that in the limit $\ell \to \infty$, all quantities (couplings, three-legs vertices, etc.) become then evaluated at temperature $T$ which is here essentially the zero temperature limit.}

\medskip

c) Fig. 8 is discussed in the text before Fig. 7. Both figures show results for specific points in Fig. 6.
$\to$ Merging the two Figures into one might be an option for better readability and a more direct view of the underlying results.

\medskip

\comment{We have interchanged the order of Figs 7 and 8 to match their discussion in the text. }

\bigskip

xviii) Figure 11: I find the legends to be a little small.

\medskip

\modif{The legend size is now larger.}

\bigskip

xix) Section 3.2 - third paragraph - "... we can write $N(\xi)=1/(\pi v_F)$": Since $v_F$ is renormalised, c.f. comment xiv), shouldn't this be the renormalised value? This would then differ somewhat from g-ology, wouldn't it?

\medskip

\comment{See comment to viii) above.}

\bigskip

xx) Section 3.2.1 - last paragraph: "... in the second quadrant": It was not clear to me which one is the "second". I'd suggest to use "lower left", "lower right", etc., to avoid ambiguities. Or it might be an option to add thin lines to separate the quadrants and to label them explicitly as e.g. I, II, III and IV. The mutually distinct nature of the physics would justify that, in my opinion.

\medskip

\modif{We thank the referee for this suggestion. The terminology "upper-left corner", etc. has been used instead of the former one.}

\bigskip

xxi) A general question and a mere matter of interest: With increasing temperature I would expect the flow to become regularised at some stage, and this should define a "one-loop $T_c$", i.e. a physical "cross-over/binding/short-range-ordering" temperature. Has this been looked at in this context?

\medskip

\comment{In line with the point xvii-b above, if the flow is integrated up to $\ell \to \infty$ at a given $T$, all quantities become evaluated at $T$. A divergence at $T$ would then signal a one-loop `critical' temperature for an instability in the charge and/or spin degrees of freedom. The corresponding `$T_c$' would be however of the same order as the gaps obtained in the $T\to 0$ evaluation used here (a factor of the order of 2 would link $\Lambda_c$ and $T_c$.). If we are interested in the nature of ground states, the $T\to 0$ procedure is significantly faster numerically. This is so because only one integration over $\ell$ is carried out, whereas as a function of $T$, an infinite integration over $\ell$ is performed at each $T$ until $T_c$ is reached. }

\bigskip

{xxii) Section 3.2.3 - page 27 lower part: "Calculations carried out ... namely up to $\mu=\sqrt{2}$ (3/4 filling)": Why is this value chosen as the upper limit for the calculations? What would happen beyond? Is the method still applicable then? If not why not? Or would it simply go beyond the scope of this work? C.f. comment xxiii) $ \to $ quarter filling.}

xxiii) It is mentioned that various previous results could be confirmed. The new results are then presented stating that "We have also carried out ... away from half filling": To my personal taste it would well be worth emphasising a bit more that these are new results, by which unexplored terrain is being entered. These are in my view the most striking results contained in the manuscript, drawing their solidity of course from the fact that the method is in line with previous results.
That said, there is some previous work on the special case of quarter filling, e.g. DOI 10.1103/PhysRevB.75.113103, and likely/possibly others (I did not manage to do a comprehensive research on this). If possible and sensible, it might be useful to compare against such prior results, too. It would be interesting if and how quarter filling might emerge as a special case within the formalism presented here. That can of course also be left for future work.

\medskip

{\color{blue} {\bf Authors comments and modifications:} The above two remarks on the applicability of the method at high doping level, namely at 3/4-filling and beyond are quite pertinent. There are no limitations of the method to carry out calculations there. Our initial reticence to show the results at and very close to 3/4 (or 1/4) filling was simply that for these we didn't include the influence of the 8$k_F$ umklapp scattering that appears as a higher commensurability effect (footnote comment of pages 27-28 in the manuscript). However, this coupling is strongly irrelevant and it does not introduce any new phase at weak coupling, but only at sufficiently strong repulsive $V$ where the onset of a charge ordered state is found, as discussed by many people and in particular by Sano and Ono in the paper quoted by the referee in the point xxiii above. Therefore for completeness, we have decided to include additional results and figures at larger dopings at the end of Section 3.3.2 in our revised manuscript. The results turn out to be rather remarkable. Sufficiently beyond 3/4 filling, for instance, a triplet phase emerges in the phase diagram in the repulsive and sizeable $V$ region. This is completely unexpected from the point of view of continuum g-ology theory, but has been found by exact diagonalisations at larger positive $V$ at 1/4 filling - e.g., Refs.~[49] and [48] of the revised manuscript - which we now discuss in connection with our results. This unexpected phase, which was left essentially without any real explanation as to both its nature and origin in the above last two references, finds an explanation from the fRG.}

\medskip

{\color{blue} {\bf Authors comments and modifications:} The work mentioned by the referee (as well as many others in the past) examines the possibility of $4k_F$ charge ordering (Wigner crystallization) at quarter filling. This ordering is a direct consequence of quarter-filling many-particle umklapp scattering that can only become relevant when repulsive $V$ is greater than some threshold value. However, the threshold falls outside the weak coupling region considered in our work. We nevertheless cite this work in the revised version alongside the one of Mila and Zotos in [50].
Finally, exact diagonalisations have also been carried out by Sano and Ono at $n=2/3$ in [48], which corresponds on hole side to $\mu=1$, that is almost identical to the case considered in the submitted version of our work ($\mu=1.03$ in Fig.~14 of the submitted manuscript). These numerical results are now compared with ours in Sec. 3.2.2, notably for the contraction of the spin gap region of the phase diagram, in sharp contrast to the g-ology results. This is well captured by the counter-screening effects found in the flow of fRG couplings. In order to make the comparison the most precise we replace the phase diagram at $\mu =1.03$, by a new one carried out at $\mu =1.0$ in a new version of Figure~14. }
\newline

\bigskip

xxiv) Some words on the specifics of the numerical implementation and the computational costs would be of interest, also to have an idea of what is possible and feasible in such a set-up. Ideally, the underlying code might be worth being developed further and even be published as a result in its own right.

\medskip

\comment{ From a numerical perspective, the algorithms used are not new, and their implementation rely on standard Python libraries devoted to the integration of differential equations.}

\medskip

\modif{Some details about the numerical method have been provided in the new appendix C.}

\bigskip

xxv) As part of the outlook, anisotropic systems are mentioned. These systems may actually permit to compare the extension of typical 1d (f)RG schemes to extensions of typical 2d (f)RG schemes, the two approaches often being quite different in nature. I wonder how the authors would feel about this aspect.

\medskip

\comment{ Recent extensions from 1D to quasi-1D - Wilsonian and fRG - use the patch momentum procedure in the determination of the three momentum variables flow of interaction vertices, which are restricted to local g-ology interactions in the parallel direction (e.g. Nickel {\it et al,} PRB {\bf 73}, 165126 (2006); C. Bourbonnais and A. Sedeki, PRB, {\bf 80}, 085105 (2009); {\bf 85}, 165129 (2012)). These also share many common aspects with the extensively developed fRG approaches to the 2D Hubbard model with the local $U$ term. However, to take into account non local interactions, like for the EHM on a quasi-1D or a 2D lattice, the number of momentum variables required for the vertices in the patch procedure would be equal to 6 rather than 3 (in order to include the effect of the lattice via irrelevant terms). The number of flow equations would then become extremely large and their solution quite demanding numerically. Extending the approach developed in our work to the quasi-1D EHM becomes appealing since the number of couplings (classified as marginal and irrelevant) would remain relatively modest. Such extension to the quasi-1D situation would be {\it per se} worthwhile since it can be directly connected to real quasi-1D materials (e.g. organic conductors) and compared with more isotropic cases like the cuprates. }

\bigskip

xxv) Appendix A.3, c.f. also comment iv): There are two parameters 'a', but for two different purposes. Also, the actual values that are used in the numerics are not specified (or I did not find them). I suggest to explicitly name them as two separate parameters and to provide the numerical values that were used.

\medskip

\modif{A different name for the two parameters has been used and their numerical values have been provided in the text.}

\section*{Reply to the second referee}

(1) The authors state in the introduction that 'These RG results were strictly speaking limited to the EHM model at half-filling.' Does this mean that traditional RG methods were not applicable when deviating from half-filling, or that they could be applied but did not yield good results? Clarifying these points and further highlighting the technical contributions developed in this paper would enhance the understanding of its significance.

\medskip

\comment{This statement was referring to the previous RG works (patch-fRG [28] and Wilsonian RG [29]) which focused on the EHM at half-filling where most of the numerical simulations have been done. There is no question that these approaches, which already went beyond those known in the continuum limit, would be applicable and controlled away from half-filling. However, in the quest of an analytical RG method - rather than numerical, as for the patch-fRG - that can classify with a low numerical cost the corrections to the continuum limit, the 1PI fRG approach, when compared to the momentum shell Wilsonian scheme, is found to be more easily implemented technically for an asymmetric tight-binding spectrum away from half-filling.}

\medskip

\modif{The paragraph at the end of the introduction on page 2 stating the goal and the main motivations of our work has been rephrased to better highlight the novelty of the RG formulation proposed. }

\bigskip

(2) This may be a trivial question on Fig.1: Why is the density of states not symmetric with respect to the band center ($\epsilon=0$) for the tight-binding model? Is there already some form of interaction incorporated in a circular manner?

\medskip

\comment{ There was an apparent error (due to a lack of resolution near the van Hove singularities) in the plot of the density of states. }

\medskip

\modif{The figure has been corrected accordingly.}

\bigskip

(3) It is unclear if the flow equations Eqs.(49-51) for channels at $q=\pm 2 k_F$ are still applicable to the case away from half filling.

\medskip

\comment{The equations are valid at arbitrary filling, and for $q$ in the vicinity of $ \pm 2 k_F $. It might be that we did not fully understand the point raised by the referee. See also (5).}

\bigskip

(4) While the qualitative agreement with previous works at half-filling is acknowledged, it is not clear if the results at half filling are also improved by the presented fRG approach?

\medskip

\comment{The results of our fRG approach at half-filling are essentially similar to previous results (e.g., [28] and [29]), the difference being marginal. They equally well stand the comparison with numerical results. }

\bigskip

(5) At small doping levels, how can it be determined that the charge gapless CDW, BOW, and SDW phases are incommensurate? It is presumed that charge and/or spin gapped phases are commensurate, but can commensurate CDW or SDW phases still occur when the system deviates from half-filling? Moreover, while the concept of incommensurate CDW and SDW is comprehensible, what constitutes an incommensurate BOW state because BOW state is characterized by a two-site unit cell?

\medskip

\comment{The bond (site) susceptibilities correspond to response to inter-site electron transfer (local) densities, which remain distinct at the wave vector $\pm 2k_F (\ne \pi)$ for a finite range of doping away from half-filling where the influence of umklapp is still perceptible at short distance (see also comment to point (6) below). At sufficiently large doping this difference vanishes. All the staggered density-wave susceptibilities are evaluated at $q=2k_F$, since this momentum corresponds to the strongest singularities (the Peierls loop $\sim \mathcal{L}_P$ is still logarithmically singular at $2k_F$). Away from half-filling, $2k_F\ne \pi= G/2$, so that the gapless and the gapped CDW, BOW, and SDW phases are incommensurate. The susceptibilities associated with a commensurate DW are weaker, since $2k_F^c=\pi$ is no longer the best nesting vector. }

\bigskip

(6) Related to question (5), when investigating the EHM lattice model using numerical methods such as DMRG or QMC, it is anticipated that away from half-filling, the charge gap in the repulsive region would close immediately. To put it plainly, at $V=0$, the Bethe Ansatz indicates that only at $n=1$ is there a charge gap due to a singularity, but it is evident that deviating from $n=1$ leads to a charge gapless state. However, Fig. 9 still shows a charge gapped state for $U>1.5$. How can this discrepancy be understood?

\medskip

\comment{It is entirely true that the Mott insulating gap immediately closes when the band filling departs from $n=1$, namely when the system is no longer an insulator but a metal with a Drude peak in conductivity whatever small is the excess or lack of carrier concentration from $n=1$. This is known in the Hubbard limit but must hold as well when $V$ is present. However, near $n=1$ a finite energy gap linked to the energy distance between the lower and - e.g. doped - upper Hubbard like sub-bands persists and does not close immediately at $n\ne 1$. This gap remains meaningful for a finite interval of doping away from half-filling. The gap found by the one-loop RG at $n\ne 1$ refers to that energy, which does not imply an insulating behavior although a one-loop RG procedure cannot really tell. This gap is an essential ingredient of the Luther-Emery description of the doped Mott insulator at $n\ne 1$ (e.g., [6], [40] and [4]). We took care to avoid calling it a Mott insulating gap. Some authors refer to it as a commensurability gap (e.g., [6] and [40]); we prefer to keep the appellation of a charge gap at $n\ne 1$ since it is occurring in the charge sector. }

\medskip

\modif{We have added a footnote on page 13 which emphasizes this important distinction. }

\bigskip

It would be helpful to the general reader if the paper could provide a clearer explanation of how the gapfull and gapless states in the charge and spin sectors are determined.

\medskip

\modif{A general remark about the identification of spin and charge gaps from the singularities of the coupling RG equations is given after Eqs. (32) in the manuscript.}

\bigskip

(8) Some figures on flow of the vertices and couplings are missing titles on the vertical axis. It would be beneficial for clarity and completeness if these could be added.

\medskip

\modif{Thank you for this remark. The missing titles have been added.}

\bigskip

(9) The boundary between the SDW and TS phases is indicated to be in the second quadrant, while the boundary between the CDW and SS phases is stated to be in the fourth quadrant. Could there be a possibility that this is opposite?

\medskip

\modif{Maybe the terminology 'first/second... quadrants' is misleading. it has been replaced with the terminology 'upper-left... quadrants'.}

\bigskip

(10) The 1D doped EHM has been studied by an fRG method in a relatively recent paper: Y.-Y. Xiang et al., J. Phys.: Condens. Matter 31 (2019). A brief statement about discrepancy to this paper may be useful.

\medskip

\comment{We thank the referee for drawing to our attention the above reference that was unknown to us. The authors use a particular approximation scheme for the momentum dependence of vertices which differs from a development in terms of marginal and irrelevant couplings, as used in our work.
Some results do agree with ours but many differ. The approximation fails for instance to recover the known g-ology results as the couplings tend to zero, although in our opinion it should. This article will be nevertheless included in the references.}

\bigskip

(11) Please correct some typos, e.g., abbreviation EHM'istwicedefined,TL' is defined in the second appearance of Tomonaga-Luttinger, there is no vector notation (arrow) in $|n|\neq 0$, etc.

\medskip

\modif{We went over our manuscript and correct these. }

---

## Editorial Decision

published